# Current Paradigms and Future Challenges in Harnessing Nanocellulose for Advanced Applications in Tissue Engineering: A Critical State-of-the-Art Review for Biomedicine

**DOI:** 10.3390/ijms26041449

**Published:** 2025-02-09

**Authors:** Mudasir A. Dar, Rongrong Xie, Jun Liu, Shehbaz Ali, Kiran D. Pawar, I Made Sudiana, Jianzhong Sun

**Affiliations:** 1Biofuels Institute, School of the Environment and Safety Engineering, Jiangsu University, Zhenjiang 212013, China; muddar7@ujs.edu.cn (M.A.D.); junliu115142@hotmail.com (J.L.); shahbaz205@ujs.edu.cn (S.A.); 2Department of Zoology, Savitribai Phule Pune University, Ganeshkhind, Pune 411007, India; 3School of Nanoscience and Biotechnology, Shivaji University, Vidyanagar, Kolhapur 416004, India; kdp.snst@unishivaji.ac.in; 4Research Center for Applied Microbiology, National Research and Innovation Agency (BRIN), Jl. Raya Jakarta-Bogor KM. 46, KST Soekarno, Cibinong, Bogor 16911, Indonesia; imad003@brin.go.id

**Keywords:** tunable properties, nanocellulose biomaterials, cytocompatibility, biomedical applications, skin burns, antimicrobial activity

## Abstract

Nanocellulose-based biomaterials are at the forefront of biomedicine, presenting innovative solutions to longstanding challenges in tissue engineering and wound repair. These advanced materials demonstrate enhanced mechanical properties and improved biocompatibility while allowing for precise tuning of drug release profiles. Recent progress in the design, fabrication, and characterization of these biomaterials underscores their transformative potential in biomedicine. Researchers are employing strategic methodologies to investigate and characterize the structure and functionality of nanocellulose in tissue engineering and wound repair. In tissue engineering, nanocellulose-based scaffolds offer transformative opportunities to replicate the complexities of native tissues, facilitating the study of drug effects on the metabolism, vascularization, and cellular behavior in engineered liver, adipose, and tumor models. Concurrently, nanocellulose has gained recognition as an advanced wound dressing material, leveraging its ability to deliver therapeutic agents via precise topical, transdermal, and systemic pathways while simultaneously promoting cellular proliferation and tissue regeneration. The inherent transparency of nanocellulose provides a unique advantage, enabling real-time monitoring of wound healing progress. Despite these advancements, significant challenges remain in the large-scale production, reproducibility, and commercial viability of nanocellulose-based biomaterials. This review not only underscores these hurdles but also outlines strategic directions for future research, including the need for bioengineering of nanocellulose-based wound dressings with scalable production and the incorporation of novel functionalities for clinical translation. By addressing these key challenges, nanocellulose has the potential to redefine biomedical material design and offer transformative solutions for unmet clinical needs in tissue engineering and beyond.

## 1. Introduction

Tissue engineering has emerged as a promising field at the intersection of biology, materials science, and engineering, offering innovative solutions for tissue repair and regeneration [1,2]. Central to the success of tissue engineering approaches is the development of biomaterials that mimic the extracellular matrix (ECM) and support cellular proliferation, differentiation, and tissue remodeling. Among these biomaterials, nanocellulose has garnered significant attention due to its unique properties, including biocompatibility, biodegradability, mechanical strength, and versatile surface chemistry [3,4]. Nanocellulose is a compelling, sustainable, non-toxic, and renewable biomaterial [5,6]. Nanocellulose derived from renewable resources such as wood pulp, bacteria, or plants represents the tiniest form of cellulose, having nanoscale dimensions [7]. Nanocellulose broadly refers to different forms, viz., cellulosic nanofibers (CNFs), cellulose nanocrystals (CNCs), bacterial nanocellulose (BNC), etc., that possess high aspect ratios and surface areas. These cellulose-based nanostructures can be modified chemically with functional groups or grafted with biomolecules. Attributed to the presence of free hydroxyl groups on surfaces, nanocellulose in all its forms can undergo chemical modification by incorporating a range of functional groups or attaching biomolecules through grafting processes. This process of functionalization enhances not only the inherent physicochemical and mechanical characteristics of nanocellulose but also introduces additional functionalities, thereby enabling customization for various specific biomedical applications [8,9]. Its nanostructured morphology provides a high surface-to-volume ratio, facilitating interactions with cells and biological molecules. The functionalization of nanocellulose improves its physical as well as chemical characteristics with controlled biological interactions, allowing the material to be tailored for a variety of applications with high specificity [10,11]. Consequently, nanocellulose has emerged as a promising candidate for various biomedical applications, particularly in tissue engineering and wound healing.

In recent years, numerous studies have explored the potential of nanocellulose in tissue regeneration and wound healing [12,13]. In tissue engineering, the nanofibrous structure of nanocelluloses like BNC resembles the native extracellular matrix, promoting cell adhesion, proliferation, and tissue regeneration [14]. Their high mechanical strength, porosity, and ability to be functionalized further enhance their suitability as scaffolds for hard- and soft-tissue engineering applications. As drug carriers, nanocelluloses enable targeted and controlled drug release due to their high surface area and tunable surface chemistry [15]. This targeted delivery enhances treatment efficacy while minimizing side effects. Nanocellulose-based wound dressings maintain a moist environment conducive to cell proliferation and accelerate wound healing. Their antimicrobial properties, achieved through surface modification, further aid in preventing infections [16]. This review highlights the current state of knowledge related to the bioengineering of cellulose-based nanomaterials and their applications in biomedical research, with a special emphasis on tissue engineering and wound healing. Here, we provide a comprehensive overview of the unique features of nanocellulose and the fabrication strategies required to improve its physicochemical characteristics and interaction with cells and tissues. Further, this work elucidates the mechanisms underlying the beneficial effects of nanocellulose on tissue engineering and wound healing while also highlighting the challenges and opportunities for future research and clinical translation. Harnessing the unique properties of nanocellulose holds tremendous potential for advancing regenerative medicine and improving patient recovery in the treatment of various diseases and injuries.

## 2. Methodology and Bibliometric Analysis

To compile this study, we performed a comprehensive literature survey on the Web of Science core collection, PubMed, ScienceDirect, and Google Scholar databases by entering keywords such as “nanocellulose AND tissue engineering”, “Nanocellulose AND wound healing”, and “nanocellulose AND skin tissue repair”. To enhance the transparency and scientific rigor of our systematic review, we adhered to the PRISMA protocol, as illustrated in Figure 1. The search returned 1075 research articles published during the last two decades between 2007 and 2024 (Figure 2). The term “nanocellulose” alone retrieved 12,988 entries, while the phrases “nanocellulose and tissue engineering” and “nanocellulose and wound healing” retrieved over 1008 and 440 entries, respectively. In other words, this indicates that these simple searches imply that comprehensive investigations of nanocellulose for wound repair are underrepresented when compared with the overall research on nanocellulosic materials. However, the publications showed a continuously increased trend after 2013, and the highest number of studies was published in the year 2023, accounting for 114 articles. The first publication highlighting the potential applications of nanocellulose for wound healing was reported by Kramer et al. [17], where they designed an innovative BC composite using specific combinations of acrylic and methacrylate monomers and varied crosslinkers, achieving significant improvements in water absorption, strength, and elasticity. The materials retain essential BC features such as shape, nanofiber network, pore system, and biocompatibility, making them ideal for collagen-like biomedical applications. Similarly, Branski et al. [18] are among the pioneers who developed a porcine wound-healing model using nanocellulose dressings. They revealed that the nanocellulose-based dressings effectively prevented infections and hematomas and enhanced graft adherence, thereby highlighting the potential of nanocellulose biomaterials in improving wound healing outcomes.

The VOS viewer software was employed along with an examination of pertinent keywords given by the authors. The findings revealed that 11 terms represented the minimum frequency (5) of keyword occurrences. Following this, a network map illustrating the terms frequently used in this research is presented in Figure 3. The connections between the circles are represented by lines, with color and size denoting the cluster category and frequency of each specific keyword, respectively. The analysis indicates that these articles predominantly emphasize three aspects: Cluster 1 highlights five items such as cellulose, animals, biocompatible materials, mice, and scanning electron microscopy (green cluster). Likewise, cluster 2 demonstrates five items, including nanocellulose, tissue engineering, wound healing, hydrogels, and drug delivery (brown cluster), while cluster 3 shows four keywords, such as nanofibers, tissue scaffolds, bioinks, and 3D printing (blue cluster). These keywords underscore the potential uses of nanocellulose biomaterials for tissue engineering and wound healing in animals, including humans (Figure 3A). In addition, these key aspects highlight the hotspots of future research on the bioengineering of nanocellulose for tissue repair (Figure 3B). Figure 3C depicts strong links between nanocelluloses and tissue engineering and wound healing by using them as hydrogels or drug-delivery agents. The bibliometric analysis performed in the VOSviewer software (Version 1.6.20) further indicates that the top 3 countries that have contributed the most to the subject area are China, India, and the USA, with 45, 21, and 12 research articles, respectively, in recent years (Figure 3D).

## 3. Structural and Morphological Characterization of Nanocellulose

Nanocellulose is the tiniest form of cellulose, having nanostructures measuring less than 100 nm in size in one dimension [19]. In certain cases, they can be hundreds of nanometers or micrometers in size, especially with electrospun nanofibers (Table 1). Based on the shape and structural dynamics, the cellulose nanostructures may be classified as nanofibrils, nanofibers, nanowhiskers, nanocrystals, nanorods, and nanoballs. Nanofibrils are commonly observed as hydrogels [20] and can originate from bacterial secretion or undergo extraction from woody plants through mechanisms such as hydrolysis, oxidation, and mechanical disintegration [21]. Similarly, nanofibers are exceedingly thin, fibrous cellulosic structures with properties akin to nanofibrils, often isolated from sources like pineapple [22]. Nanofibers, although slightly thicker and longer than nanofibrils, are typically produced via electrospinning of cellulose blended with other synthetic polymers or composites [19]. Despite having diameters exceeding 100 nm, electrospun nanofibers are commonly referred to as “nanofibers” [23]. Cellulose nanowhiskers, nanocrystals, and nanorods, on the other hand, exhibit fibrous morphologies with diameters comparable to nanofibrils but are characterized by shorter lengths. Nanocrystals may assume needle-like or rod-shaped morphologies [24], while nanorods are specifically rod-shaped nanocrystals [25]. The terms “cellulose nanowhiskers”, “cellulose nanocrystals”, and “cellulose nanorods” are often used interchangeably in the scientific literature [21]. The nanostructures are derived through the selective breakdown of the amorphous regions within cellulose molecules, achieved primarily via acid hydrolysis while preserving the crystalline components (Figure 4). Figure 4 illustrates various methods, key stages, and the main technologies employed in the synthesis of nanocellulose. Consequently, the resultant CNC, nanowhiskers, and nanorods exhibit heightened crystallinity indices compared with cellulose nanofibrils. The latter are typically generated via enzymatic hydrolysis [26] or mechanical processes such as grinding and shearing [21], leading to the removal of non-crystalline regions. Moreover, cellulose nanoballs, or nanocrystalline cellulose particles with a spherical morphology, are documented entities [27]. In contrast, nanoplatelets represent plate-like structures composed of nanofibrils with thicknesses on the nanoscale, often extending to micrometer dimensions in other axes [28].

The nanofibrils can be transformed into larger two-dimensional (2D) and three-dimensional (3D) micro- and macrostructures such as membranes [29], films (2D structures), or microneedles and sponges (3D structures) [30]. Based on the recent reports, the morphological structure of each of these nanomaterials is described in Table 1. Since the shape, size, and surface composition of nanocellulose significantly affect its rheological properties in aqueous suspensions, further research is needed to establish structure–property relationships and understand how different nanocellulose morphologies influence viscosity and flow behavior, which is crucial for processing and combining with other materials [27]. Furthermore, the source (plants, algae, bacteria) and the defibrillation method used to extract the nanocellulose can result in varying structural features, such as crystallinity, porosity, and surface chemistry. More investigations are anticipated to understand how these structural differences influence the properties and performance of nanocellulose in various applications. Similarly, the quantification of the anisotropic nature of freeze-cast nanocellulose scaffolds, with varying mechanical properties for different nanocellulose types and processing conditions, is yet to be determined.

**Table 1 ijms-26-01449-t001:** List of the different types of nanocellulose and characteristics suitable for various biomedical applications.

Type of Nanocellulose	Diameter (nm)	Source/Examples	Methods Used for Preparation	Other Characteristics	Reference(s)
Nanofibrils	70–140	Bacteria, plants, algae	Bacterial secretion	High crystallinity and pure cellulose (no hemicellulose, lignin), good mechanical stability, production from low-molecular-weight molecules	[31,32]
	3–5	Softwood-derived cellulose	TEMPO oxidation and enzymatic pretreatment		[33,34]
Nanofibers	≤400	Cellulose acetate, bacterial, lignocellulose, algae, tunicates	Chemical synthesis	Gel-like characteristics in water; long, flexible fibers capable of entanglement; contain both crystalline and amorphous regions	[3,8]
	≤6.4	Pineapple	Electrospinning		[22]
Nanowhiskers	5–15	Pine kraft pulp	Acid hydrolysis		[35]
	10–15	Kenaf bast	Acid hydrolysis		[36]
	10–100	Bacteria, plants, algae	Bacterial secretion	High crystallinity and pure cellulose (no hemicellulose, lignin), good mechanical stability	[37]
Nanocrystals	≤7.3	Cotton-derived, plants	Acid hydrolysis	Elongated crystalline, rodlike shapes; rigid rods (elastic modulus ~100 GPa); crystalline; no amorphous regions; form stable hydrogels with up to 99% water	[38,39]
Nanorods	15 ± 3	Grass-derived	Acid hydrolysis		[40]
Nanoballs	80–85	Wood-derived	Acid/alkaline hydrolysis		[41]
Nanoplatelets	≤80	Agave-derived	Aqueous dispersion and heat treatment		[42,43]

## 4. Overview of the Nanocellulose Resources

In contemporary discourse, the burgeoning interest in sustainable nanocellulose production underscores the exploration of unconventional sources, notably industrial remnants from beer manufacturing plants [44] and municipal solid wastes, heralding a paradigm shift toward eco-conscious material synthesis [45]. The nanocellulose is produced from four natural sources, viz., plants, bacteria, algae, and animals (Figure 4) [19]. Plant-based nanocellulose is obtained from a diverse array of sources, such as trees, shrubs, herbs, grasses, flowers, root vegetables, succulents, etc. Trees include birch [46] and conifers, e.g., *Pinus radiata* [47]. Depending on the source, nanocellulose is also classified as either hardwood (from broad-leaved trees) or softwood (from conifers). Cotton and hibiscus are examples of shrub-derived nanocellulose [48]. Agricultural leftovers are also proven as valuable sources of nanocellulose. Materials like sugarcane bagasse [49], *Miscanthus giganteus* [50] or *Imperata brasiliensis* [51], bamboo [52], rice husk, corn leaf, triticale straw, pineapple leaf, soybean straw [26], and agave [28] have been explored and may offer promising avenues for the extraction of nanocellulose. Furthermore, unconventional sources like pineapple leaves, soybean straw, carrots, and agave are being investigated for their nanocellulose potential. This widening exploration of diverse biological materials reflects a move toward more sustainable and versatile nanocellulose production methods.

Bacterial cellulose, also known as microbial cellulose, is synthesized extracellularly by various Gram-negative bacterial genera, including *Acetobacter*, *Achromobacter*, *Aerobacter*, *Agrobacterium*, *Alkaligenes*, *Azotobacter*, *Pseudomonas*, *Rhizobium*, *Rhodobacter*, *Salmonella*, and *Sarcina* [53]. Among bacteria, the bacterium *Gluconacetobacter xylinus* is the most widely utilized bacterial species for nanocellulose synthesis [54]. However, other bacterial species, including *G. hansenii* [55], *G. kombuchae* [56], *K. europaeus*, and the low-pH-resistant strain *K. medellinensis* [57] have also demonstrated nanocellulose production. The addition of yeast extract in the culture media [58], as well as symbiotic co-cultivation with yeasts or *Medusomyces gisevii* [59], are known to boost bacterial growth and nanocellulose synthesis. Structurally, bacterial cellulose shows a distinctive reticulate network of fine fibers akin in composition to plant cellulose, albeit devoid of lignin, pectin, and hemicelluloses [60].

Noteworthy, algae are an attractive and sustainable source for the production of nanocellulose, offering several advantages over conventional plant-based sources. Algae represent a sustainable and promising source for the production of nanocellulose with unique properties and diverse applications, contributing to the development of eco-friendly and high-performance materials. Among algae, *Cladophora* [30] and *Cystoseria myrica* [61] serve as notable sources of nanocellulose. Nanocellulose from *Cladophora* species is highly crystalline, making it more chemically inert and resistant to degradation. Renowned for their abundance and biocompatibility, algal-derived nanocellulose finds extensive utility in various domains, particularly in biomedical applications and environmental remediation efforts. Algal-derived nanocellulose has been evaluated as a scaffold for the enhancement of cell cultivation [62], hemocompatibility [63], and dye adsorption capabilities [30]. It is also frequently used for the removal of contaminants such as heavy metals, glucans, and endotoxins [64]. Additionally, the nanocellulose obtained from *C. Myrica,* coupled with Fe_3_O_4_, has been used to reduce mercury pollution [61].

Nanocellulose acquired from animals is pivotal for industry and biotechnology. Animal sources of nanocellulose are relatively limited compared with plant and bacterial sources. Among animals, tunicates like *Styela clava* (Herdman, 1881) [65] and *Halocynthia roretzi* (Drasche, 1884) [66] are widely known to produce significant amounts of cellulose that can be extracted and used for nanocellulose production. The cellulose content in tunicate tissues can be as high as 88% by weight, making them a rich source of cellulose [67]. The cellulose extracted from tunicates exhibits unique properties, such as high crystallinity and purity, making it suitable for producing nanocellulose, particularly CNC [68]. Nanocellulose derived from tunicates has been explored for various applications, including water treatment, biomedical engineering, and the development of nanocomposites [69]. Cellulose films produced from *S. clava* have been investigated for wound dressings [65]. They might also be used for sewing fibers, tissue-engineering scaffolds, absorbable hemostats, and hemodialysis membranes [70]. The nanocellulose membrane extracted from *Halocynthia roretzi* has been used for the removal of oils from wastewater [71]. Zhao and Li investigated the tunicate-derived cellulose from four different species: *Ciona intestinalis* (Linnaeus, 1767), *Ascidia* sp. (Linnaeus, 1767), *Halocynthia roretzi*, and *Styela plicata* (Lesueur, 1823) [72]. The authors compared the quantity, purity, and structural characteristics of the cellulose extracted from these species and revealed that hard tunics obtained from *Halocynthia roretzi* and *Styela plicata* yielded more cellulose than other species, with 100% purity and 95% crystallinity. While tunicates represent a unique animal source of nanocellulose, their availability and scalability for industrial production may be limited compared with plant-based or bacterial sources. Nevertheless, the unique properties of tunicate-derived nanocellulose make it an interesting material for further research and prospective application in biomedicine.

### Synthesis of Nanocellulose: Bacterial vs. Chemical Synthesis

During the last decade, several researchers have pioneered diverse methodologies for the synthesis of nanocellulose, delineating its distinct attributes tailored for a broad spectrum of applications [53]. Nanocellulose can be prepared from wood or plant biomass by using mechanical, enzymatic, or chemical methods (Figure 4), wherein the lignin and hemicellulose components are selectively removed before the deconstruction of the cellulose’s hierarchical architecture [73]. Notably, the 2,2,6,6-tetra methyl piperidine-1-oxyl radical (TEMPO)-mediated oxidation method, pioneered by Isogai and his collaborators in 2011, has emerged as the predominant approach for extracting CNFs with diameters ranging from 5 to 60 nm and lengths spanning several microns [74]. Conversely, CNCs are obtained by subjecting the amorphous regions of cellulose pulp to robust acid hydrolysis while preserving the integrity of the crystalline domains within the fibrils [3]. Generally, the extracted CNC is 5 nm in diameter and 20–100 nm long. However, the sizes of CNCs and CNFs vary considerably, contingent on the source material and the complexities of the involved treatment processes.

Many bacteria are known to synthesize nanocellulose, called BNC [2,75]. The BNC is secreted by bacterial strains of *G. xylinus*, *Komagataeibacter*, *Agrobacterium*, *Salmonella*, *Aerobacter*, *Rhizobium*, *Escherichia*, *Sarcina*, and *Rhodobactor* [76]. The average diameter and length of BNC are 100 nm and several micrometers, respectively. BNC is usually secreted as an extracellular product of fermentation when the bacteria are grown in a culture medium containing glucose as a substrate [77]. Washing the bacterial cellulose with 1 M NaOH helps to remove the unwanted proteins and obtain pure BNC. This nanocellulose possesses higher crystallinity (80%) and remains free of contaminants like lignin and hemicellulose observed in wood-derived nanocellulose [78]. Being relatively clean and environment friendly, BNC possesses properties superior to chemically synthesized nanocellulose, such as high crystallinity, purity, and mechanical strength [79]. Despite bacterial synthesis producing highly pure and crystalline BNC through a relatively clean process, chemical synthesis offers greater versatility, scalability, and cost-effectiveness for producing nanocellulose from various biomass sources, albeit with the potential for environmental concerns due to chemical usage [80,81].

## 5. Physicochemical Properties of Nanocellulose from the Tissue-Engineering Perspective

### 5.1. Structures and Physical Properties of the Nanocellulose Biomaterials

Cellulose-based aerogels represent highly porous nanostructured materials with several distinct advantages (Figure 5). The unique features of cellulose-based aerogels include mechanical robustness, a high degree of polymerization, exceptional purity, and crystallinity, rendering them promising candidates for applications with resistance to higher pressure [82]. The CNC-based aerogels exhibit a spherical morphology, contrasting markedly with the rice-shaped configuration of CNF-based aerogels, attributed chiefly to the elongated filamentous nature of CNC. The mechanical properties of aerogels hinge upon two pivotal parameters: the choice of precursor material and the complexity of the preparation methods [83]. Cellulose, renowned for its reinforcing capabilities, augments the mechanical integrity of numerous hybrid aerogels. Moreover, the chemical modification of cellulose serves to further enhance its mechanical resilience. Manipulation of the structural attributes of cellulose aerogels is relatively facile, offering avenues for tailored applications [84]. For biosensing applications, Edwards et al. fabricated nanocellulose-based aerogels boasting 99% porosity, incorporating peptides for the detection of protease enzyme activity [85]. The mass spectral analysis and the physical features of the regenerated aerogels were found acceptable for the treatment of wounds, showing intelligent protease sequestration.

### 5.2. Cytocompatibility of Nanocellulose

To date, nanocellulose has been amalgamated with various biological constituents such as proteins, glycosides, growth factors, cytokines, and other polysaccharides, among others, to amplify its therapeutic efficacy (Table 2). The unique physical, chemical, and mechanical characteristics of cellulose-based aerogels, as well as their biocompatibility, make them suitable for a diverse array of biomedical applications (Figure 6). Hence, their biocompatibility, in addition to several structural characteristics, facilitates their integration with other natural materials, thereby fostering the creation of green composites having novel features [86]. During fabrication, various antibacterial agents are immobilized and incorporated within the nanocellulose matrices, endowing them with antibacterial properties. Uddin and coworkers, for instance, immobilized silver nanoparticles (AgNPs) and antibacterial peptides inside CNFs [87]. The authors asserted that CNF serves as a highly effective substrate for bioactive substances, facilitating the preservation and sustenance of enzymatic and antibacterial functionalities. Lu et al. [88] generated dialdehyde CNF/collagen composite aerogels with excellent porosities of 90–95%, with high water absorption capacity (up to 4000%) and biocompatibility. The enhanced water absorption capability inherent in aerogels makes them suitable for wound dressings. Recently, bacterial cellulose has been combined with chitosan to augment the mechanical properties and antibacterial effects of wound dressings [89]. The integration of medical-grade diamond nanoparticles served to augment the mechanical characteristics of cellulose composites derived from BNC and chitosan. The inclusion of these nanoparticles resulted in a diminution in the diameter of the nanocellulose fibers, facilitating the electrospinning procedure. Additionally, these nanodiamond-modified mats were more hydrophilic, thereby demonstrating their applicability for the proliferation and adhesion of mouse skin fibroblasts, and thus, rendering them potential biomaterials for skin tissue engineering [90]. Sericin, a major constituent of silk produced by silkworms, is a key protein in the modification of bacterial cellulose [91]. A sericin-releasing BNC gel was created as a bioactive mask for facial therapy. The diffusion of sericin from the bacterial cellulose boosted fibroblast proliferation, enhanced cell viability, and improved the synthesis of extracellular matrix. Consequently, bacterial cellulose/silk sericin composites have emerged as a promising material for application in wound treatment and tissue engineering [92]. In pursuit of expediting the re-epithelialization process, smart membranes have been designed from oxidized bacterial cellulose fused with epidermal growth factor (EGF). The controlled release of EGFs is induced through the lysozyme, a ubiquitous enzyme abundantly present at sites of infected skin wounds [93]. Macrophage-stimulating protein (MSP), a pivotal cytokine produced in adipose-derived mesenchymal stem cells (ASCs), is also used with BNC owing to its wound-healing properties. Upon amalgamation with BNC and administration onto a skin wound, MSP expedited the wound healing process, likely attributable to the migration of dermal fibroblasts toward the MSP, thereby augmenting collagen production [94]. The most important factor for an effective wound dressing lies in its ability to protect the wound against microbial infections caused by fungi and bacteria. Despite its reputation as an ideal wound dressing, BNC inherently lacks antibacterial activity when applied alone. Consequently, ongoing research endeavors are focused on exploring the compatibility of BNC with antibacterial agents such as metals, antibiotics, antiseptics, and other naturally occurring antimicrobial molecules. Among antiseptics, notable examples include povidone-iodine and polyhexamethylene biguanide (PHMB) along with octenidine [95,96], which have been frequently integrated into Poloxamer micelles for subsequent injection into BNC matrices, thereby facilitating the prolonged release of octenidine [97]. The clinical investigation evaluated the efficacy of BNC-based wound dressings containing the antibacterial PHMB [98]. Recent advancements have highlighted the efficacy of metal-based agents, such as silver sulfadiazine and silver nanoparticles, in combating *Pseudomonas aeruginosa*, *Escherichia coli*, and *Staphylococcus aureus* within nanocellulose wound dressings [53,99]. Similarly, gold nanoparticles (AuNPs) modified with 4,6-diamino-2-pyrimidinethiol are incorporated into the bacterial cellulose for antimicrobial activities [100]. These composites have demonstrated significant inhibitory effects on several Gram-negative bacterial strains.

Hitherto, many studies have revealed that the immobilization of antibiotic dosages inside nanocellulose aerogels increases the efficiency of the compounds [151]. The unique cross-linking properties and surface chemistry of cellulose enable it to mix with many biomolecules, including protein extracts, antibiotics, and even metal nanoparticles [151]. Shen et al. prepared hybrid nanocellulose-based aerogels with AgNPs as antibacterial agents [133]. The authors partly destroyed the crystalline regions of the cellulose and allowed oxidation to increase the absorption of nanoparticles by the hybrid gels. Upon testing, they observed strong antibacterial activity against a variety of pathogenic bacteria. Similarly, Ye incorporated amoxicillin into cellulose aerogels for controlled release of the antibiotic compound in order to enhance its performance [134]. Other antimicrobial compounds like quaternary ammonium are also incorporated into bacterial cellulose for antipathogenic activities against *S. aureus* and *S. epidermidis* [152]. Ceftriaxone, a third-generation cephalosporin, is an example of an antibiotic used in combination with nanocellulose for wound healing [135]. One of the naturally existing antibacterial compounds, chitosan, is mixed with BNC for bacteriostatic functions against *E. coli* and *S. aureus* [153]. Furthermore, lignin and lignin-derived compounds [136], and curcumin, a naturally occurring polyphenolic molecule identified from *Curcuma longa*, are other natural antimicrobial molecules. Due to low solubility, curcumin is applied in combination with plant-derived and chemically modified nanocelluloses only for wound dressings. It is rarely used with bacterial cellulose. Curcumin showed improved wound healing and antibacterial properties when entrapped in a composite made of gelatin and ionically modified self-assembled bacterial cellulose [154]. Bacakova and colleagues also observed that bacterial cellulose loaded with the degradation products of curcumin reduced the number of *S. epidermis* colonies [19]. Another emerging approach to fighting bacterial infections in wounds is the delivery of antibacterial peptides such as nisin, a polycyclic antibacterial peptide generated by the bacterium *Lactococcus lactis*. Nisin is incorporated via the electrostatic interaction between the negatively charged TEMPO-oxidized nanofibrillar cellulose surface and the positively charged peptide molecules, showing enhanced activity against *Bacillus subtilis* and *S. aureus* compared with free nisin [117]. The composites containing Alkannin and Shikonin were also found to have antibacterial properties against *Propionibacterium acnes*, suggesting their significance in treating acne [155].

Other drugs that can be incorporated into nanocellulose include anticancer medicines such as α-mangostin, which reduce the development of B16F10 melanoma cells and MCF-7 breast cancer cells [118]. In addition to antibacterial activity, Alkannin, Shikonin, and their derivatives possess wound healing capacities as well as antibacterial, anti-inflammatory, antioxidant, and anticancer potentials. These fascinating compounds are also integrated into electrospun cellulose acetate nanofibrous meshes for prospective wound dressings [156]. Vaccarin, a flavonoid glycoside famous for stimulating neovascularization, when combined with BNC membranes improved the re-epithelialization of skin wounds in rats [54]. Lidocaine has been applied with bacterial cellulose as a localized anesthetic to relieve pain and enhance wound healing, particularly in the case of burns [149]. Another method used to deliver lidocaine onto wound sites is based on biodegradable microneedles prepared from bacterial cellulose and fish-derived collagen. The addition of glycerin to bacterial cellulose is one of the promising strategies for wound recovery. Glycerin provides a moisturizing effect that might be useful in the treatment of skin illnesses that are characterized by dryness, such as psoriasis and atopic dermatitis [145]. The BNC in the form of nanocrystals is usually preferred for this therapy. Moreover, the bacterial CNC is also employed to fortify regenerated chitin fibers, making them applicable to skin sutures [150]. Bromelain, a protease found in pineapple tissues that shows anti-inflammatory and anticancer activities, is also incorporated into nanocellulose [115]. However, much research is needed to explore the specific pathways and mechanisms that govern the immune responses for the biocompatibility of NC within biological systems.

### 5.3. Biodegradability of Nanocellulose-Based Biomaterials

In humans and many other vertebrates, cellulose may not be degraded easily due to the absence of cellulase enzymes; therefore, nanocellulose represents an excellent material for tissue engineering in these animals. Nevertheless, structural attributes such as crystallinity, immune response, and absorption may affect the rate of degradation in vivo. The studies related to in vivo biodegradation of nanocellulose-based materials inside the human body are relatively few, and it needs further exploration, although some scientists have attempted to investigate the biodegradability of nanocellulose-based tissues in many animals. Solhi and coworkers observed that the hydrolysis of cellulose or its derivatives in canines is highly dependent on the crystallinity and the method of extraction [157]. Deacetylated cellulose, having a highly crystalline structure, did not degrade over 6 weeks. In contrast, the amorphous cellulose was easily absorbed, showing a 75% degradation in 6 weeks. Moreover, under moist conditions or aqueous environments, CNCs are degraded faster than carbon nanotubes and fullerenes; however, the in vivo confirmation of these tests remains to be verified [158]. Some authors have inferred that oxidized cellulose is more susceptible to biodegradation, signifying its potential degradation by the human body. Consequently, scientists are continuously trying to improve the biodegradability of nanocellulose through oxidative processes [159]. The authors further characterized the TEMPO-oxidized nanocellulose for potential applications in tissue engineering. The authors observed that TEMPO-mediated oxidation significantly improved the in vitro degradation of bacterial cellulose in aqueous environments, such as phosphate-buffered saline and simulated body fluid, by periodate oxidation [159]. The periodate oxidation of gamma-irradiated bacterial cellulose occurs in two major phases. Primarily, the sample undergoes rapid degradation, hydrolyzing about 70–80% of the sample, followed by a slower degradation rate of 5–10% that eventually levels off, leaving the residual non-resorbable materials only. These observations were also supported by the in vivo degradation of oxidized BNC in white rabbits, which revealed continuous degradation of the cellulosic membranes, with the most frequent degradation taking place in the first 4 weeks [160]. For tissues like artificial heart valves and menisci, a highly biocompatible but non-biodegradable structure is preferred. In the case of other tissues, for example, artificial bone grafts, bioresorbable materials are always a good choice [108,161]. Future research focused on elucidating the primary routes and rates of NC degradation within the bodies of animals is anticipated.

### 5.4. Immunogenicity of Nanocellulose

Once applied, nanocellulose materials are known to show low or no cytotoxicity and immunogenicity. The CNF obtained from the leaves of *Ananas erectifolius* are non-cytotoxic, as they demonstrated no direct or indirect signs of toxicity to the viability and morphology of Vero cells (Table 3). Moreover, the cells showed a higher affinity for the CNF surface during adhesion tests [22]. Similarly, the cotton-derived CNC is an example of non-immunogenic nanocellulose. It does not influence the production of pro-inflammatory cytokines such as tumor necrosis factor (TNF) and interleukin-1 (IL-1) by macrophages in humans [162]. However, recently many in vitro as well as in vivo studies have emerged that reported considerable cytotoxicity and pro-inflammatory activities of nanocellulose. Bhattacharya et al. [163] compared the cytotoxicity of nanocellulose with ZnO-, Ag-, and SiO_2_-based nanotubes and nanomaterials by Alamar blue assay. The authors observed that nanocellulose was practically non-cytotoxic for human macrophages like THP-1 cells. However, the cytokine and chemokine profiling revealed substantial inflammatory responses at sub-cytotoxic doses of the compound [163]. Similarly, in the case of mice, the CNC induced an inflammatory response after aspiration, showing an increased number of leukocytes and eosinophils in the lungs in addition to the up-regulation of many pro-inflammatory cytokines and chemokines, viz., TNF-a, G-CSF, GM-CSF, INF-, MCP-1, and MIP. Moreover, the nanocrystals also manifested oxidative stress and tissue damage by showing an increase in lactate dehydrogenase activity and accumulation of oxidatively damaged proteins in the bronchoalveolar fluid [164]. Shvedova et al. [163] have reported similar kinds of inferences in their investigation. The wood-derived CNC also induced pulmonary inflammation and oxidative stress along with increased collagen and transforming growth factor levels in female mice [165]. Likewise, in rats, sulfonated nanocellulose derived from *Khaya sengalensis* seeds can cause renal damage [166].

Despite limited cytotoxicity and immunogenicity, these properties of the nanocellulose can be altered by the addition of specific chemicals or an electrical charge. It has been reported that the modification of nanocellulose with carboxymethyl (anionic nanocellulose) and hydroxy propyl trimethyl ammonium groups (cationic nanocellulose) elicits a lower pro-inflammatory effect on human dermal fibroblasts, lung cells, and THP-1 macrophages [175]. The shape and structure of the particles also impact the cytotoxicity and immunogenicity of nanocellulose. The nanofibrillar cellulose shows more cytotoxicity and induces higher oxidative stress on human lung epithelial cells than CNC. However, when compared with CNF, the CNC caused a more inflammatory response with elevated pro-inflammatory cytokine and chemokine secretions. It is astonishing to note that CNC particles, but not CNF particles, are taken up easily by the cells, as revealed by cellulose staining assays [176].

Moreover, several in vivo studies have demonstrated significant immunological responses to CNF and CNC in mice. When the mice were exposed to CNF for prolonged periods, immune cell polarization showed unique patterns along with TH1-like immune reactions, whereas CNC induced non-uniform immunological responses. However, both forms of nanocellulose elicited a milder reaction than asbestos and carbon nanotubes [177]. The addition of many other organic compounds such as curcumin suppresses the cytotoxicity of CNC, eliciting a decrease in IL-1 secretion in murine macrophages [24]. Precisely, the immunogenicity of bacterial and algal nanocellulose may be due to impurities like endotoxin and 1,3-D-glucan, which can be reduced via genetic engineering of these microorganisms [178]. However, how nanocellulose induces the immune responses, and what pathways or mechanisms are involved in the process, are still elusive, and therefore demand further exploration.

### 5.5. Non-Toxic Nature of Nanocellulose

Despite the earlier studies that reported low or no toxicity of nanocellulose, the toxicological and safety concerns for these natural nanomaterials should be investigated thoroughly. Although the toxicity of nanomaterials like AgNPs, AuNPs, quantum dots, etc. follows a comprehensive assessment strategy, the toxicology of nanocellulose or nanocellulose-based biomaterials is still in its nascent stages, with few studies related to cytotoxicity only. The cytotoxicity of nanocellulose-related materials is summarized in Table 3. Overall, the nanocellulose shows insignificant impacts on the cellular and genetic machinery of the animals and their organs. Notwithstanding this, the inhalation of nanocellulose, particularly CNC, in bulk amounts may affect the pulmonary systems of animals due to its easy self-aggregation and non-degradable nature. Some studies have investigated the intrinsic ecotoxicology of CNC in several species of fish [179]. The CNC showed low environmental risks, with negligible cytotoxicity on the hepatocytes of fishes, posing no harm to aquatic life at concentrations that usually occur in the environment. In another study, the cytotoxicity of CNC was tested on nine different cell lines by using MTT and lactate dehydrogenase (LDH) assays. The authors observed no cytotoxic effects of CNC against any of the tested cell lines up to 48 h of treatment with doses ranging from 0 to 50 µg/mL [180]. However, some recent studies have found dose-dependent cytotoxic and inflammatory effects of CNC on human lung cells, posing a risk with exposure to high concentrations of CNC for longer durations [181]. Although CNFs have shown acute environmental toxicity to some bacteria in the environment, they do not demonstrate any cytotoxicity or inflammatory effects on mouse and human macrophages [182]. The researchers at the VTT Technical Research Centre in Finland have speculated that CNF demonstrates low cytotoxicity with no DNA or chromosomal damage to mice despite a slight pulmonary inflammation caused by particulate/bacteria from the nanomaterials [157]. Similarly, Pereira et al. investigated the in vitro cytotoxicity of CNF and its impact on gene expression in fibroblasts. Low concentrations of CNF (100 µg/mL) showed no evident toxicity; however, higher concentrations of 2000 and 5000 µg/mL may result in a loss of cell viability and low expression of stress- and apoptosis-related genes [183]. The CNF crosslinked with polyethyleneimine (PEI) and CTAB significantly reduced cell viability and proliferation compared with pure CNF [184]. In the same manner, the cationic modified-CNF (trimethylammonium-CNF) showed greater cytocompatibility than unmodified and anionic modified CNF [124,185]. Owing to its simple and natural synthesis, BNC is considered the most compatible biomaterial. The mouse feeding experiments revealed that it is non-toxic to the endothelial cells and osteoblasts [186]. Even though studies on nanocellulose have found no serious environmental or biological concerns, more research and systematic assessments of nanocellulose ecotoxicology are direly needed, particularly to investigate the effects and mechanisms of nanoparticle aggregation in the body in addition to long-term in vivo toxicity testing. Furthermore, not only the direct toxic effects of nanocellulose materials on tissues, but the indirect toxicity generated by the incorporation of nanocellulose, is another important area of research that might highlight the ecotoxicology of these products in the related environment.

## 6. Surface Modifications of Nanocellulose

Nanocellulose’s exceptional versatility stems from its ability to undergo tailored surface modifications, transforming it into a highly sophisticated and intelligent biomaterial for tissue engineering [187,188]. Hitherto, several reactions have been explored for the surface modification of nanocellulose, such as esterification, sulfonation, phosphorylation, cationization, silanization, initiation, adsorption, coating, oxidation, polymer grafting, etc. [189,190]. In particular, chemical modification strategy-based functionalization enables precise control over its surface chemistry. By introducing functional groups like carboxyl, hydroxyl, or amino moieties through oxidation, coupling, or coating techniques, nanocellulose achieves enhanced biocompatibility, reactivity, and targeted bioactivity [191]. In tissue engineering applications, the functionalization of nanocellulose serves as an advanced platform for drug delivery, wound healing, and scaffold engineering, where its tailored surface chemistry enables antimicrobial activity, growth factor immobilization, and controlled drug release [192]. Moreover, these modifications enhance its compatibility with biological tissues, promote cell adhesion, and improve its interaction with bioactive molecules. For instance, TEMPO-mediated oxidation generates carboxyl-functionalized nanocellulose, facilitating crosslinking with polymers and proteins, thereby improving hydrogel stability and bioactivity [193]. Esterification and etherification impart hydrophobicity and mechanical resilience, making them ideal for scaffold reinforcement, while silanization enhances the interfacial properties of nanocellulose by introducing reactive silane groups for better integration with biomolecules.

Another strategy, phosphorylation of nanocellulose, is explored for its bioactivity, making it highly relevant for bone tissue engineering applications [194]. By introducing phosphate groups onto the nanocellulose surface, this functionalization significantly increases the material’s affinity for calcium ions—an essential component of bone mineralization [191]. This property not only facilitates the formation of a mineralized matrix but also actively supports cellular adhesion, proliferation, and osteogenic differentiation, thereby improving the integration of the engineered scaffolds with the native bone tissue [195]. In the context of biomimetic bone regeneration, phosphorylated nanocellulose-based hydrogels could serve as dynamic scaffolds for calcium phosphate deposition, closely mimicking the natural biomineralization process crucial for skeletal tissue development [192]. Recent advancements have underscored well the potential of phosphorylation of nanocellulosic biomaterials [191,196]. Wang et al. demonstrated the incorporation of phosphorylated CNF into a dextran/methacrylated gelatin emulsion bioink for extrusion-based 3D bioprinting [196]. The modified bioink exhibited superior rheological behavior, enhanced damping capacity, and increased mineralization ability, leading to improved cell viability, osteogenic differentiation, and biomineralized nodule formation. In summary, the surface modification of nanocellulose highlights its transformative potential in the development of next-generation bioactive scaffolds, offering a compelling route toward advanced regenerative therapies in orthopedic and craniofacial tissue engineering. 

## 7. Biomedical Applications of Nanocellulose-Based Biomaterials

Nanocellulose-based materials exhibit tremendous characteristics that enable them to have diverse applications including tissue repairing, wound healing, and controlled drug delivery (Figure 6). Various physicochemical and biological features such as the electrical charge and wettability of the nanocellulose can be tailored by different sources and methods to meet the demands of tissue engineering and other biotechnological applications. The presence of oxygen-containing chemical functional groups (-OH) in nanocellulose molecules determines their wettability, which can be further regulated by pretreatment with oxidation during their production. This oxidation is frequently accelerated by the TEMPO, which provides –COOH groups to the nanocellulose [21]. It is well understood that a moderate wettability of the material surface causes the adsorption of cell-adhesion-mediating proteins from biological fluids in a geometrical configuration that is accessible to cell adhesion receptors and therefore improves cell adhesion and growth [197]. The nature of the charge, whether negative or positive, is an important factor for interaction with cells. The –COOH groups attribute a negative charge to the nanocellulose, which varies with the density of these molecules present in the mixture [198]. The charge density therefore influences the shape and roughness of the nanocellulose films [199] as well as their interactions with cells, impacting their growth, viability, and susceptibility to DNA transfection [46]. When compared with the effect of uncharged nanocellulose, some authors have revealed that both the negative as well as positive charges enhanced cell adhesion properties in addition to reducing the cells’ immune activation [200]. Moreover, anionic nanocellulose was shown to provide better support for cell adhesion and proliferation than cationic nanocellulose, which was made by adding ammonium groups to nanocellulose [201]. However, anionic charges produced a more significant activation of the immune response than cationic-charged nanocellulose [186].

During the last decade, nanocellulose has been applied frequently to tissue engineering and related areas such as wound healing and cell–material interactions involving various biotechnologies, such as biosensing and controlled drug delivery. As previously indicated, research on the potential use of nanocellulose in neural tissue engineering, cartilage tissue engineering, and skin wound dressings, as well as hepatic, vascular, and bone tissue engineering, has progressed with encouraging results.

### 7.1. Tissue Engineering Applications of Nanocellulose-Based Biomaterials

The first reference describing nanocellulose appeared in 2006, authored by Kramer et al. [17], who studied the biomaterial properties of nanocellulose for tissue engineering. The authors were mainly inspired by the biocompatibility, mechanical strength, and water retention properties of the nanocellulose in addition to its controllable shape, elasticity, and nanofibrous and nanoporous structures. The initial study was focused on the development of collagen-like materials made from composites of bacterial cellulose and synthetic polymers by photopolymerization of acrylate and methacrylate monomers and methacrylate crosslinkers [17]. In 2007, Bodin and colleagues studied the use of nanocellulose for tissue engineering by functionalizing BNC scaffolds with cell-adhesion-mediating GRGDS oligopeptides [202]. The in vitro testing revealed that these scaffolds significantly improved the adherence of human vascular endothelial cells, suggesting their potential use in vascular tissue engineering [202]. After 2012, there was a surge of research on the possible use of nanocellulose in tissue engineering and the interaction of nanocellulose with cell(s). Correspondingly, the first paper describing the application of nanocellulose for neural tissue engineering was published in the journal *Biomaterial Science Polymer Edition* [139]. In that investigation, different composite membranes prepared from BNC and polypyrrole (PPy) were applied as seeding templates for PC12 rat neural cells. It was observed that the cells grew better on BNC/PPy composites than on pure BNC. Furthermore, the presence of electrically conductive PPy allowed stimulation of the cells, which is important for cell functioning [139].

Nanocellulose is widely used for tissue engineering involving tissue repair and wound healing (Figure 7). Engineering of blood vessels, neural tissue, bone, cartilage, liver, and adipose tissue; reconstruction of the urethra and dura mater; repairing connective tissue and congenital heart defects; constructing protective barriers; and ophthalmologic applications, primarily contact lens construction, are some of the most common examples of these applications [203,204]. Nanocellulose has also been used to improve the efficacy of cell transfection and to create a 3D culture environment for stem cells to preserve their pluripotency [205,206]. For neural tissue engineering, the adherence, proliferation, and differentiation of SH-SY5Y neuroblastoma cells was demonstrated for the first time when cultured on 3D-BNC [207]. Cationic alteration of this material with trimethyl ammonium β-hydroxy propyl cellulose can significantly improve the adhesion, proliferation, and development of 3D neural networks on 3D BNC scaffolds, as observed on PC12 neuron cells [208]. Additionally, nanocellulose-manufactured constructs were used as an innovative tool for studying the functions of the brain. To achieve this, a bioink containing cellulose nanofibrils and carbon nanotubes was exploited to make a 3D print of electrically conductive scaffolds that stimulated the adhesion, proliferation, and differentiation of human SH-SY5Y neuroblastoma cells [209].

#### 7.1.1. Nanocellulose-Based Materials for Cartilage Engineering

The excellent water-retention capacity and mechanical strength of CNF have led to the further development of auricular cartilage restoration. Due to its high mechanical strength and a host tissue response similar to the auricular cartilage, BNC containing 17% cellulose content could serve as a viable non-resorbable biomaterial for auricular cartilage tissue engineering [210]. The bilayer scaffolds made of BNC and alginate are also promising biomaterials as they promote the development of human nasoseptal chondrocytes in addition to their non-pyrogenic and non-cytotoxic nature [210]. Moreover, in articular cartilage engineering, BNC scaffolds modified by laser perforation have been employed as substrates for culturing human chondrocytes. These unique and novel scaffolds increase the efficiency of nutrient transportation as well as the growth and differentiation of chondrocytes. Further, the deposition of the newly developed extracellular matrix is also caused by the nanocellulose scaffolds [103]. The application of nanocellulose-based bioink in 3D bioprinting with live cells is another characteristic of nanocellulose as a promising biomaterial. A similar biobased ink was used for 3D printing with irradiated human chondrocytes and induced pluripotent stem cells (iPSC) that were derived from joint cartilage [211]. Similarly, a bioink prepared from the alginate sulfate/BNC enhanced collagen II synthesis, proliferation, and overall growth in femoral condyle cartilage-derived bovine chondrocytes [74]. The double cross-linked interpenetrating polymer network prepared from the sodium alginate and gelatin hydrogels is another interesting composite biomaterial for cartilage tissue engineering [212]. It performs well when reinforced with 50% CNC. The BNC scaffolds were also used under in vitro conditions for the reconstruction of the human auricle [213] while combining with human primary chondrocytes that were collected through normal septorhinoplasties and otoplasties. Another model used was bovine knee cartilage having a punch defect filled with BNC [214].

#### 7.1.2. Nanocellulose and Hepatic Tissue Engineering

Nanocellulose has been increasingly used for other applications such as adipose tissue engineering and liver tissue engineering. The initial step in liver tissue engineering was to construct a 3D culture of hepatic cells, which is more appropriate physiologically than the 2D culture used for the prediction and estimation of drug metabolism, excretion, and toxicity in the human liver. To achieve this, 3D scaffolds of nanofibrillar cellulose derived from birchwood were utilized [12,33,215]. These scaffolds improved the functions and differentiation of the HepaRG human liver progenitor cells, which were obtained from the liver tumor of a female patient having hepatitis C virus infection and hepatocarcinoma. These HepaRG cells generated 3D multicellular spheroids with apicobasal polarity and functioning bile canaliculi-like structures. These spheroids provided an increased surface area for the hepatobiliary proteins like MRP2 and MDR1 that are responsible for drug transport toward the biliary compartment. Moreover, in the HepaRG cell cultures, the 3D hydrogel supported the mRNA expression and metabolic activity of hepatocyte markers like albumin and CYP3A4 [216].

#### 7.1.3. Nanocellulose in Adipose Tissue Engineering

Recently, nanocellulose has been engineered to develop three-dimensional models of adipose tissue aimed at advancing studies in adipose biology and metabolic disorders such as obesity and diabetes [80,217]. The 3D scaffolds were prepared by crosslinking homogenized BNC fibrils with alginate and then freeze-drying them to form a porous structure. The 3D scaffolds contained more C3H10T1/2 mesenchymal stem cells with markers of adipogenic cell differentiation, such as growing in clusters and containing large lipid droplets, than did 2D BNC scaffolds in an adipogenic medium. All these properties indicate that 3D scaffolds offer significant promise, not only for in vitro investigations but also for adipose tissue reconstruction after trauma or tumor removal and congenital diseases [217]. Henriksson et al. created a similar system of 3D printing with a bioink consisting of nanocellulose and hyaluronic acid-containing adipocytes [218]. The adipocytes depicted a uniform distribution over the scaffolds, a mature phenotype, and higher viability than the cells cultured in 2D culture systems.

#### 7.1.4. Nanocellulose in Vascular Tissue Engineering

Nanocellulose scaffolds have been tested for vascular system research also. In vascular tissue engineering, tubular structures have been prepared from BNC by adopting silicone tubes as molds. These tubes have great potential to replace hollow organs like the ureter and the esophagus [219]. Weber et al. performed an in vivo replacement of the right carotid artery in sheep with BNC tubes [220]. After removal, the histological inspection revealed no acute symptoms of foreign body reaction such as giant cell immigration or other inflammatory reactions, thereby indicating the biocompatible nature of the BNC tubes. However, the tubes were prone to thrombotic blockage, necessitating antiplatelet medication during implantation [220]. Another intriguing study covered endovascular stents with BNC and superparamagnetic iron oxide nanoparticles, which attracted vascular smooth muscle cells (VSMCs) for in situ development of tunica media in blood vessels [198,221]. Some in vitro experiments revealed that the magnetic BNC coated with polyethylene glycol manifested excellent scaffolds for porcine VSMCs, exhibiting little cytotoxicity. Further, these scaffolds showed beneficial effects on cell survival and migration. These materials also demonstrated good mechanical qualities and could serve as a promising matter to treat brain vascular aneurysms [222,223]. Some BNC scaffolds, after functionalization with IKVAV peptides, were studied for vasculogenic mimicry of human melanoma SK-MEL-28 cells. These scaffolds also appeared as promising 3D platforms for anti-tumor drug screening [55]. 

#### 7.1.5. Nanocellulose in Bone Engineering

Bacteria-secreted nanocellulose shows considerable potential for bone tissue engineering even in its unaltered condition. The BNC is known to enhance the multilayered growth, adhesion, and osteogenic differentiation of bone marrow mesenchymal stem cells (MSCs) in rats [224,225]. The MSCs on BNC scaffolds formed a mature type of collagen I and exhibited alkaline phosphatase activity, which was demonstrated via second harmonic generation imaging technology [226]. Moreover, the composite of nanofibrillar cellulose and chitin has proved a promising material for the construction of scaffolds for bone tissue engineering, as observed with human MSCs. The performance of nanocellulose-based bone-forming cells like rat calvarial osteoblasts and MSCs could be further improved with biomimetic mineralization through calcium phosphate, hydroxyapatite, and tricalcium phosphates [227,228]. These scaffolds can also be used in conjunction with collagen I or osteogenic growth peptide [56]. Nanocellulose has also shown promise as a coating for bone implants. For a strong bone-to-implant contact and accelerated healing process, a hybrid coating containing 45S5 bioactive glass was individually wrapped and interwoven with fibrous CNC followed by deposition on stainless steel. The nanocellulose-based coating promoted the adhesion, spreading, proliferation, and differentiation of the mouse MC3T3-E1 osteoblast progenitor cells in addition to mineralization of the extracellular matrix formed by these cells under laboratory conditions [229]. Similarly, the coating of 3D-printed polycaprolactone scaffolds with hydrophilic nanofibrillar cellulose stimulated the adhesion, differentiation, and proliferation of MSCs derived from human bone marrow [230]. Nanocellulose can also be used for the treatment of intervertebral disc degeneration [231]. The gellan gum hydrogels supplemented with BNC have been developed as substrates for regenerating the outer part of the disc, i.e., annulus fibrosus [183].

#### 7.1.6. Application of Nanocellulose for Urethra Reconstruction

In addition to all the above features, in rabbit and dog models, 3D porous bacterial cellulose scaffolds seeded with rabbit lingual keratinocytes and a microporous network of silk fibroin were used, respectively, for the reconstruction of the urethra [232]. To this end, the authors of [233] pre-seeded the keratinocytes and smooth muscle cells of dogs with nanocellulose-based bilayer scaffolds. The nanoporous network astonishingly supported the epithelial cells, whereas the microporous scaffolds augmented the growth and penetration of smooth muscle cells. Another study evaluated the use of a 3D porous BC scaffold for urethral reconstruction in rabbits. The BC scaffold supported cell infiltration and neovascularization, leading to the regeneration of urethra-like tubular structures [234]. The porous structure of the BC scaffold facilitated cell migration, proliferation, and tissue ingrowth, which are crucial for urethral tissue regeneration. Nanocellulose-based scaffolds can be functionalized with bioactive molecules or combined with other biomaterials to enhance their properties for urethral tissue engineering applications [12,232].

### 7.2. Engineering of Nanocellulosic Biomaterials for Other Tissues

The other fascinating applications of nanocellulose include the repair of connective tissue, congenital heart defects, cell transfection, and ophthalmology. To test the repair of connective tissue, softwood pulp-derived CNC was injected into skin and tendon specimens obtained from pigs and subject to stretch injuries by mechanical testing equipment [235,236]. These matrices were reinforced mechanically due to nanocellulose treatment, as evidenced by their higher elasticity and mechanical strength. At the same time, no cytotoxicity was observed with rat primary patella tendon fibroblasts. Furthermore, the activity of mitochondrial enzymes was increased significantly in the cells cultured for 2–3 weeks in the presence of CNC compared with the control experiments [237]. Nanocellulose was also used for the manufacture of contact lenses [204,238]. Nanocellulose formed a clear and very transparent macroporous hydrogel when mixed with poly (vinyl alcohol) and 90% water. In addition to high transparency, the refractive index of the hydrogel was almost near that of water, showing excellent UV-blocking characteristics and elasticity of typical soft tissues [239]. Bacterial cellulose membrane is an ideal material to prevent the leakage of cerebrospinal fluid, thus acting as a prospective dural patch for the reconstruction of the dura mater. The leakage of the cerebrospinal fluid is a common problem, frequently observed after cranial and spinal surgeries. The membranes further supported the viability and attachment of dural fibroblasts in humans [240]. Furthermore, cellulose “neurotubes” caused the accumulation of neurotrophic factors, facilitating neuron regeneration [241].

Nanocellulose’s structure and electrical charge density can also alter the efficacy of cell transfection. To date, several nanofibrillar celluloses in the form of films or hydrogels have been prepared from birch kraft pulp with low or high charge densities [242,243]. The nanocellulose-based films with low charge densities demonstrated higher efficacy for the transfection of HeLa cells than the films with high charge densities and hydrogels with both low and high charge densities [199]. Furthermore, matrices with low charge densities aided in the growth, survival, and proliferation of encapsulated HeLa cells and T lymphocytes [46]. Moreover, BNC scaffolds were shown to be effective substrates for the attachment of human umbilical vein endothelial cells and mouse MSCs after conjugation with fibronectin and type I collagen [244]. Other types of nanocellulose, such as nanowhiskers or nanocrystals, as well as nanofibrils of BNC, have been demonstrated to offer tremendous promise in tissue engineering and other biomedical applications [245].

The tissue engineering of nanocellulose presents several challenges that researchers must address to realize its full potential. Key challenges include ensuring biocompatibility to avoid adverse immune responses and understanding the long-term effects of nanocellulose within biological systems. Tailoring the mechanical properties of nanocellulose to match specific tissue requirements and maintaining the structural integrity of the scaffolds is crucial. Additionally, controlling the degradation rate to align with tissue regeneration, understanding the metabolic pathways of degradation products, and developing scalable, cost-effective production methods with consistent quality are significant hurdles. The functionalization of nanocellulose surfaces to improve cell attachment, proliferation, and differentiation without compromising material properties is another critical area of focus. Understanding cellular responses and immune interactions with nanocellulose in different tissue environments is fundamental. Researchers must explore how to control and tune the degradation rate to match tissue regeneration and ensure the safety of the degradation products. Investigating how mechanical properties can be engineered to mimic various tissues and the impact of mechanical loading on scaffold integrity is crucial. Effective functionalization strategies to promote cell attachment and growth without compromising biocompatibility and mechanical properties are essential areas of study. Additionally, understanding the in vivo performance of nanocellulose compared with in vitro studies, the long-term outcomes of implanted scaffolds, and the potential synergies from integrating nanocellulose with other biomaterials is important. Addressing these challenges and questions will pave the way for the successful integration of nanocellulose in tissue engineering, potentially transforming the field and opening new avenues for regenerative medicine.

### 7.3. Wound Healing and Skin Tissue Repair Applications of Nanocellulose

Skin represents a remarkably complex and multifunctional organ that serves as the primary interface between an organism and its external environment. Its complex architecture, comprising a stratified epithelium (epidermis) and an underlying connective tissue layer (dermis), confers a multitude of vital physiological roles. Firstly, the skin acts as an exquisitely structured physicochemical barrier, protecting against a myriad of environmental effects, including microbial pathogens, noxious chemical agents, and deleterious electromagnetic radiation, particularly in the ultraviolet spectrum. Additionally, it safeguards thermoregulation and against mechanical trauma and minimizes transepidermal water loss, thereby maintaining hydroelectrolytic homeostasis, in addition to secreting some vital compounds that include pigmentary compounds, vitamin D precursors, and structural proteins like keratins, which are instrumental in the formation and maintenance of epidermal appendages such as hair and nails [246]. Therefore, it is critical to regenerate or at the very least repair injured skin, notably through skin tissue engineering and wound healing processes. Nanocellulose provides various advantages for these applications, including adequate mechanical strength and high water-absorption capacity, which allow it to preserve moisture in injured skin while absorbing the exudate from wounds [247]. The nanoscale morphology of nanocellulose mimics the architecture of the native extracellular matrix, making it a suitable substrate for skin cell adhesion and growth (Figure 8). Many in vivo studies have confirmed the healing of skin defects in mice by BNC-based wound dressings [248]. Table 4 highlights the diverse potential of nanocellulose-based composites in wound healing and skin tissue repair.

Skin tissue engineering involves reconstructing the epidermis and dermis tissues. Since bacterial cellulose resembles natural soft tissues in many aspects, it is widely used for the regeneration of the epidermis and dermis in wounded skin [246]. Bacterial cellulose is a nanofibril-containing hydrogel network that resembles the fibrillar component of a natural extracellular matrix. The application of cellulose in skin regeneration was invented long before the introduction of the term “nanocellulose”, when it was simply called bacterial cellulose. Though it is a hydrogel containing cellulose nanofibrils, bacterial cellulose pellicles were proposed as “temporary skin replacements” for healing burns, ulcers, abrasions, and other skin ailments as early as 1990 [260]. In 2006, human-transformed skin keratinocyte and human normal skin fibroblast cell lines were grown on thin films of bacterial cellulose as substrates [261]. Keratinocytes were able to develop, proliferate, and migrate on the films, whereas fibroblasts formed clusters and detached from them. This was explained by the comparatively poor cell–material adhesion compared with the relatively robust cell–cell adhesion of fibroblasts, creating contractile forces [261]. However, Kingkaew et al. stated that bacterial cellulose films were suitable substrates for the adhesion, dissemination, and development of human skin keratinocytes as well as fibroblasts [262]. Similarly, a surface-structured 3D network of bacterial cellulose nanofibers supported human keratinocytes and fibroblasts and accelerated wound healing in mice [263].

Combining bacterial cellulose with other biologically active compounds can boost the adherence and development of skin cells. Enriching bacterial cellulose films with chitosan, for example, increased the adherence of human keratinocytes to these films [262]. Human skin keratinocytes of the HS2 cell line and human skin fibroblasts had better attachment, proliferation, and morphology after incorporating keratin from human hair into the bacterial cellulose [264]. In addition, many authors have stated that the adherence and proliferation of human keratinocytes from the HaCaT line, as well as their penetration into the scaffolds to a depth of 300 μm, were supported well by composites regenerated from bacterial cellulose and gelatin. Gelatin scaffolds had a higher wound closure effectiveness (93%) than pure bacterial cellulose scaffolds (63%) within in vivo mouse models [265]. Materials like polypyrrole and polyaniline, which are electroactive composites of bacterial cellulose, and conducting polymers also hold promising potential for skin tissue engineering [259]. There is a plethora of information available about the prospective applications of bacterial cellulose in skin regeneration and tissue engineering [31,147]. Using paraffin microspheres, novel microporous 3D scaffolds with controlled pore sizes were created from BNC that augmented the proliferation of embryonic NIH 3T3 fibroblasts in mice [116].

Achieving the use of nanocellulose as a biomaterial for skin tissue repair involves several key scientific challenges. Ensuring biocompatibility and avoiding cytotoxicity are primary concerns requiring safe concentration levels and long-term safety studies. Tailoring mechanical properties to match the native skin’s elasticity and strength while maintaining durability is crucial. Promoting integration with native tissues to enhance healing processes like re-epithelialization and collagen deposition is essential. Functionalization to improve bioactivity and enable the controlled release of therapeutic agents adds complexity. Controlling the degradation rate to match the healing timeline and understanding the safe clearance of degradation products are necessary. Manufacturing challenges include developing scalable, reproducible methods that ensure consistent quality and cost-effectiveness. Navigating regulatory pathways and conducting rigorous trials are critical for clinical approval and implementation. Addressing these challenges through interdisciplinary collaboration is vital for advancing nanocellulose-based skin tissue repair.

### 7.4. Biosafety Considerations of Nanocellulose-Based Biomaterials

Although nanocellulose exhibits promising biocompatibility and non-toxicity to tissues, comprehensive biosafety evaluations are necessary before its widespread adoption in tissue engineering. A primary concern is the presence of endogenous and exogenous impurities, including residual lignin, hemicellulose, (1,3)-β-D-glucan, heavy metals, and bacterial endotoxins, which can provoke cytotoxic, immunogenic, or pro-inflammatory responses if not meticulously removed [178,266,267,268,269]. Endotoxin contamination, particularly in BNC, necessitates stringent purification strategies such as alkali treatments, enzymatic detoxification, and high-temperature sterilization to meet pharmacopeial standards [270]. Although some in vitro and in vivo studies broadly support nanocellulose’s biocompatibility, surface modifications including cationization can induce cytotoxic or genotoxic effects, necessitating precise control over functionalization [271]. Its interaction with the immune system and blood components further underscores the importance of hemocompatibility assessments, particularly in implantable and injectable applications [272]. Despite the promise of nanocellulose-based biomaterials for tissue engineering, the lack of standardized safety protocols across global regulatory frameworks remains a bottleneck for their clinical adoption [267]. Thus, future research must prioritize the development of universally accepted purification methodologies, long-term in vivo toxicity assessments, and mechanistic insights into nanocellulose’s biological fate to unlock its full potential as a next-generation biomaterial in regenerative medicine.

## 8. Challenges of Nanocellulose-Based Biomaterials in Tissue Engineering


Despite its promising properties, several challenges limit the application of nanocellulose in wound healing and tissue engineering. A key limitation is its inherent non-degradability in the human body, which can lead to scar formation or prolonged tissue retention [273]. Strategies such as enzymatic degradation using cellulases, incorporation of N-acetylglucosamine residues via metabolic engineering of *Gluconacetobacter xylinus*, and chemical modifications like oxidation have shown potential to improve its biodegradability [274]. However, achieving precise degradation rates synchronized with tissue regeneration remains challenging, as mismatched degradation can result in inflammatory responses or scaffold failure.

Bioactive functionalization of nanocellulose is another critical challenge. While its biocompatibility and adaptability facilitate the incorporation of antimicrobial agents and bioactive molecules, advanced functionalization techniques such as conjugation with peptides or growth factors require further optimization to enhance therapeutic efficacy. Similarly, developing stimuli-responsive composites for controlled drug release and scaffold degradation could significantly improve clinical outcomes. Lastly, bridging the gap between laboratory research and clinical translation poses another significant challenge. Well-designed preclinical and clinical trials are essential to validate the safety, efficacy, and scalability of nanocellulose for specific medical applications. Moreover, developing sustainable and eco-friendly production methods for nanocellulose by utilizing renewable feedstocks and minimizing environmental impact will be critical for its large-scale adoption in biomedical applications.

## 9. Future Perspectives

While nanocellulose hydrogels have shown biocompatibility for 3D cell cultures, the enzymatic digestion required for their removal raises concerns about long-term effects on cells. Furthermore, their application in culturing entire human organs remains unachieved, limiting studies to tumor cells and stem cell lines. In wound healing, nanocellulose gels require combination with other biopolymers for 3D bioprinting, while the incorporation of antibiotics or bioactive agents is essential for enhanced regeneration and infection control [31]. Future research should explore advanced functionalization strategies, including bio-conjugation with growth factors or peptides and the development of stimuli-responsive nanocellulose composites for controlled drug delivery and scaffold degradation. Additionally, optimizing nanocellulose biosensing capabilities and exploring novel designs for analyte purification or biomolecule selection could expand its utility in diagnostics and therapeutic applications.

Commercialization of nanocellulose-based materials faces hurdles such as high production costs, low yield, and energy-intensive processes. Leveraging low-cost substrates like agro-industrial waste and optimizing scale-up processes can enhance cost-efficiency and market viability [275]. Moreover, the escalating threat of antibiotic resistance necessitates the exploration of antibacterial nanoparticles, probiotics, and other novel therapies despite challenges like aggregation, cytotoxicity, and environmental persistence [27]. To fully harness the potential of nanocellulose, future efforts must focus on developing multifunctional and defect-free composites that combine affordability, safety, and robust physicochemical properties. Comparative studies of CNC, CNF, and BNC will also be critical for tailoring applications to specific biomedical needs. Strategic interdisciplinary research and innovation will be instrumental in advancing nanocellulose-based wound dressings and tissue engineering scaffolds to clinical use.

## 10. Conclusions

Nanocellulose-based biomaterials possess remarkable wound healing properties, including the ability to maintain a moist wound environment, superior biocompatibility, and robust mechanical strength. However, their intrinsic lack of antimicrobial activity poses a limitation in preventing infections during wound healing. To address this, several antimicrobial agents such as metal and metal oxide nanoparticles, antibiotics, natural and synthetic polymers, and bioactive molecules are often integrated into nanocellulose to impart antibacterial properties. These enriched nanocellulose-based wound dressings exhibit significant therapeutic potential, effectively reducing secondary infections, accelerating healing, and maintaining high biocompatibility and biodegradability with minimal toxicity. Nevertheless, challenges such as antibiotic resistance, the potential toxicity of metal oxides, the instability of natural antimicrobial agents, and environmental concerns associated with these modifications must be carefully managed to ensure sustainable and safe applications. Beyond wound healing, nanocellulose is increasingly employed in tissue engineering, with promising advancements in liver, adipose, vascular, bone, and cartilage tissue applications. Future research is anticipated to focus on the development of hybrid nanocellulose-based materials, including combinations of organic/inorganic, natural/synthetic, and diverse nanomaterials to create innovative antibacterial wound dressings and scaffolds with enhanced functionality.

## Figures and Tables

**Figure 1 ijms-26-01449-f001:**
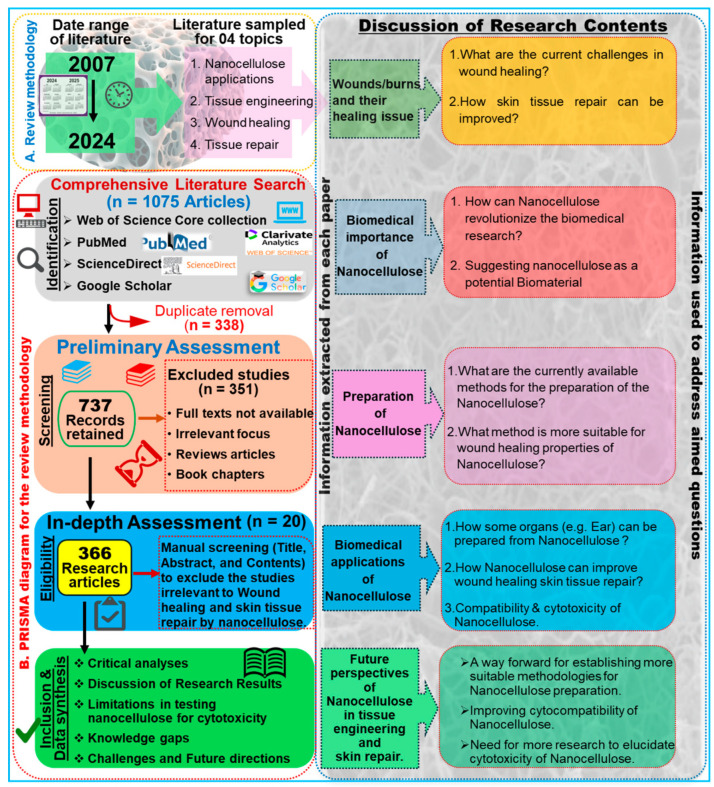
Overview of the review methodology: A. Outline of the literature review design, article collection process, and formulation of research questions. B. Flowchart illustrating the PRISMA approach, detailing the acceptance and rejection criteria for publications. Based on the relevant inquiries, goals, and objectives, a total of 366 articles were identified as eligible from an initial pool of 1075 articles.

**Figure 2 ijms-26-01449-f002:**
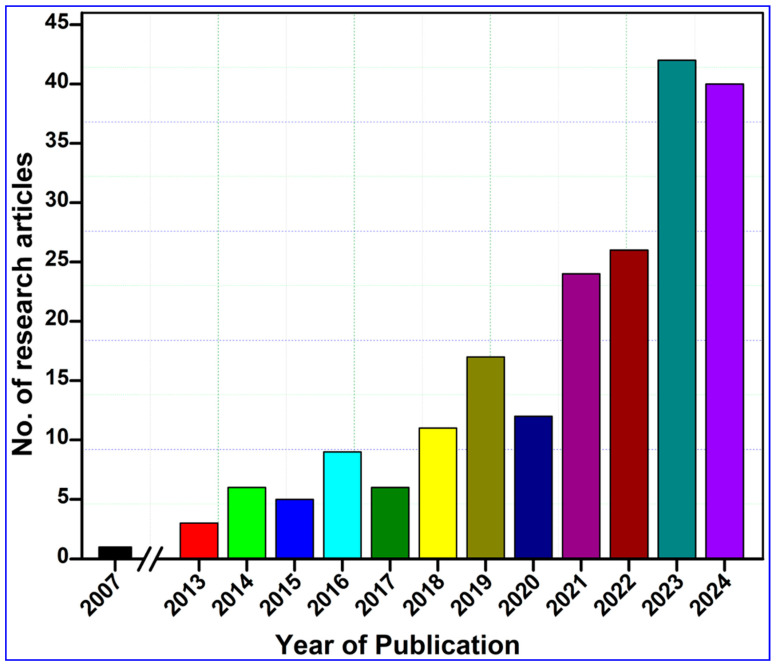
Year-wise publication records on the topic, “Tissue engineering and wound-healing properties of nanocellulose”. The related data were retrieved from the Web of Science, PubMed, ScienceDirect, and Google Scholar databases published up to 31 December 2024 by searching key phrases, such as “nanocellulose” and “wound healing” and “tissue engineering” and “skin tissue repair” in the title/abstract/author keywords of the reported research.

**Figure 3 ijms-26-01449-f003:**
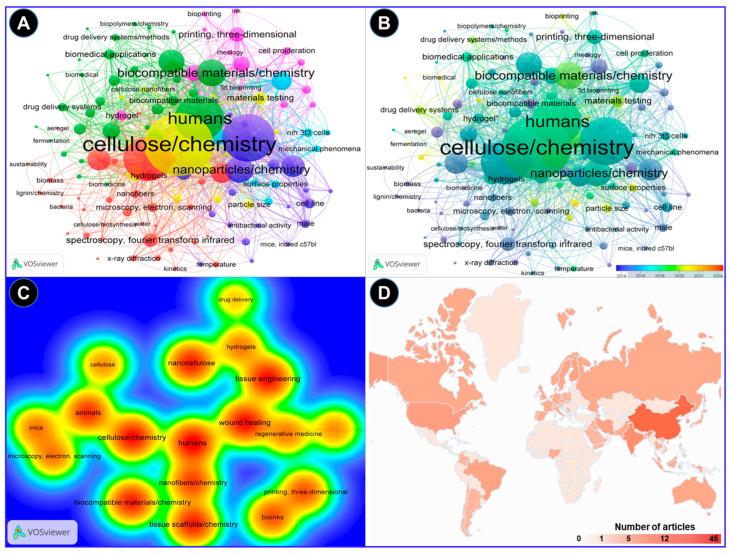
A bibliometric analysis of the reported research on applications of nanocellulose for wound healing and tissue engineering. The analysis was performed in the VOSviewer software. (**A**) A keyword co-occurrence network analysis indicating the correlation of the nanocellulose with possible biomedical applications in humans. (**B**) The overlay visualization depicted the change in the scientific focus in recent years toward the wound repair mechanisms of nanocellulose dressing. (**C**) The density visualization of nanocellulose research analyzed by the full counting method suggests close links of nanocellulose with tissue engineering and wound healing. (**D**) Country-wise distribution of the research on the topic based on the number of research articles published.

**Figure 4 ijms-26-01449-f004:**
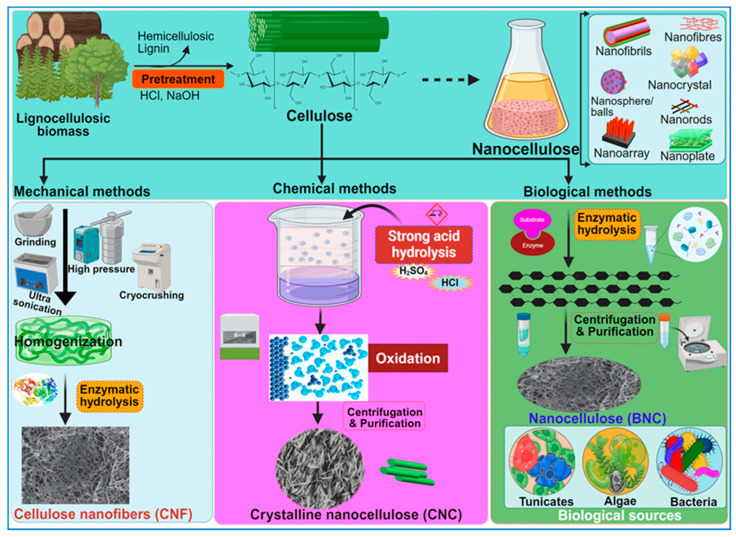
A schematic overview of the methods used for the synthesis of nanocellulose. The diagram illustrates key stages in the production process, highlighting techniques and procedures involved in deriving nanocellulose from raw materials.

**Figure 5 ijms-26-01449-f005:**
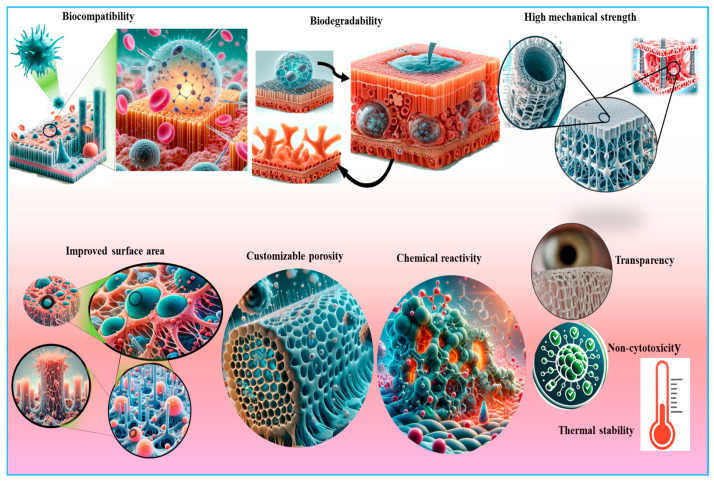
An illustration of the unique properties of nanocellulose that make it suitable for various biomedical applications, including tissue engineering, wound healing, and skin tissue repair. Key attributes include biocompatibility, high surface area, mechanical strength, and the ability to promote cell adhesion and proliferation.

**Figure 6 ijms-26-01449-f006:**
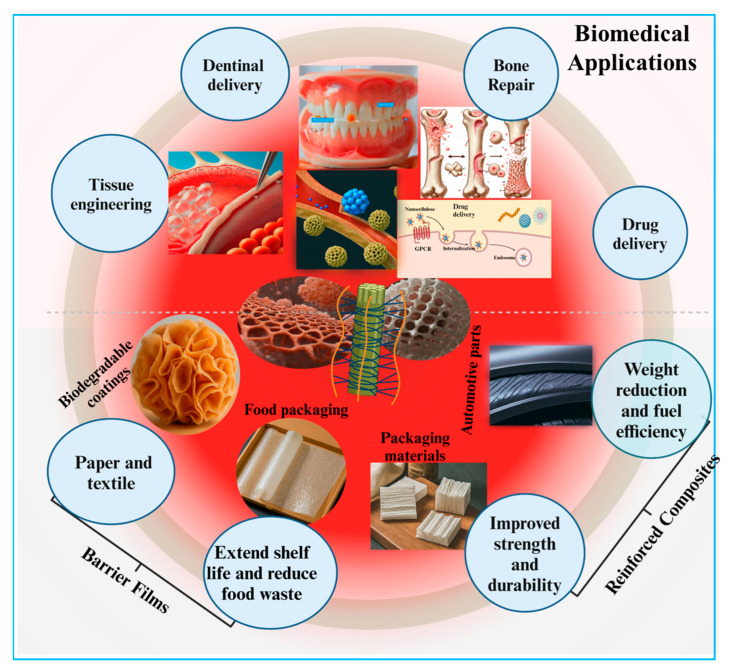
Applications of nanocellulose-based biomaterials emphasizing their biomedical potential.

**Figure 7 ijms-26-01449-f007:**
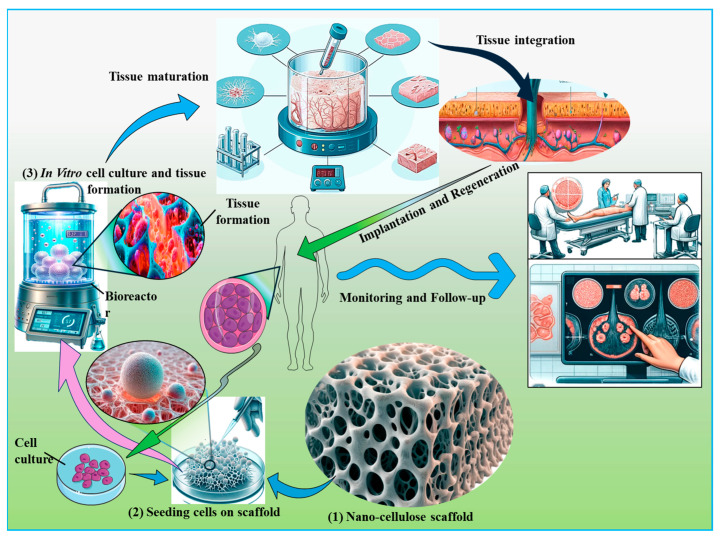
Integration of nanocellulose in tissue engineering applications. This figure illustrates the role of nanocellulose in supporting cell growth and differentiation within tissue scaffolds. The scaffold is depicted as integrating seamlessly with native tissues, thereby promoting regeneration and functional recovery. Key stages in the process are highlighted, including scaffold preparation, cell seeding, and subsequent tissue integration, emphasizing the potential of nanocellulose to enhance tissue engineering outcomes through improved biocompatibility and structural support.

**Figure 8 ijms-26-01449-f008:**
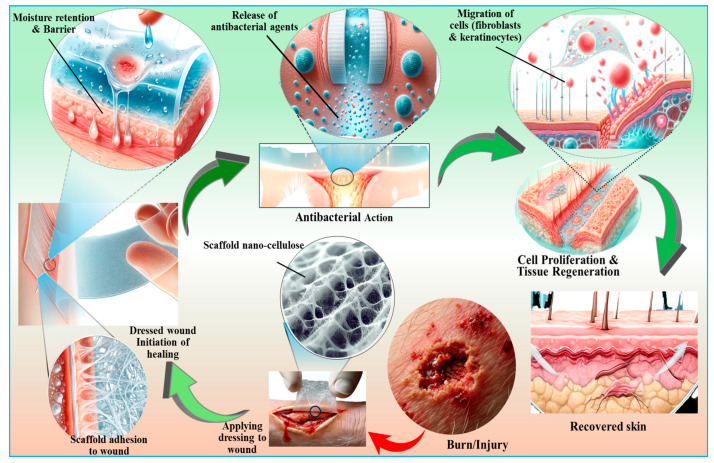
An overview of the biomedical applications of nanocellulose-based biomaterials with respect to skin repair and wound healing perspectives. Nanocellulose encourages the binding and multiplication of cells by acting as a tissue bioscaffold. It offers precise drug delivery capabilities and possesses antimicrobial properties, further underscoring its versatility in advancing biomedical technologies.

**Table 2 ijms-26-01449-t002:** Cytocompatibility of nanocellulose with different compounds with importance for skin tissue repair and wound healing.

Nanocellulose Type	Biomedical Application	Biocompatible Molecule	Nature of the Compound	Reference(s)
CNCs	Wound dressing	Elastase tripeptide	Biosensor	[31]
	Tissue engineering	Propranolol hydrochloride	Drug	[12]
		Theophylline	Stimulant	[101]
	Diagnostics	Bovine serum albumin	Protein	[102]
	Skin repair/tissue engineering	Collagen, hyaluronan	Growth factor	[103,104]
	Tissue engineering	Paclitaxel, docetaxel, etoposide	Anticancer drug	[105]
	Skin repair/wound healing	Procaine hydrochloride	Local anesthetics	[106]
	Tissue engineering	Doxorubicin hydrochloride	Anticancer drug	[107]
		Riboflavin	Vitamin	[108]
	Wound healing	Lysozyme	Enzyme	[109]
	Tissue engineering	Peroxidase	Enzyme	[110]
		Alcohol oxidase	Enzyme	[111]
		Glucose oxidase	Enzyme	[112]
		Tryptophan-based peptides	Protein	[113]
		Human neutrophil elastase	Enzyme	[114]
	Diagnostics	Papain enzyme	Enzyme	[1,115]
	Tissue engineering			
		Heptapeptide	Protein	[116]
		Diblock protein (Elastin-co-Cartilage oligomeric matrix)	Drug	[116]
	Wound healing	Nisin	Antibacterial peptide	[117]
	Tissue engineering	Anticancer drugs	Biochemical	[118]
CNFs	Skin repair	Alkonnin and Shikonin	Antibacterial agent	[119,120]
	Tissue engineering	Indomethacin, nadolol, atenolol, metoprolol tartrate, verapamil, ibuprofen	Drug	[31]
		Paracetamol	Drug	[121]
		Caffeine	Stimulant	[122]
	Wound healing	Itraconazole	Drug	[123]
		Lysozyme	Enzyme	
		Indomethacin	Drug	[124]
		Itraconazole	Antifungal agent	
	Tissue engineering	Beclomethasone dipropionate	Drug	
		Alkaline phosphatase; anti-hydrocortisone antibody	Enzyme	[125]
	Wound healing	Avidins	Protein	[126]
	Tissue engineering	Pancreatic serine protease trypsin	Enzyme	[127]
		Antihuman IgG antibody	Biomolecule	[128]
		Human immunoglobulin G	Biomolecule	[129]
		Lipase	Enzyme	[130]
	Wound healing	AgNPs	Metal and antibiotic	[87]
	Tissue engineering	Collagen	Protein	[88]
	Wound healing/tissue engineering	Chitosan, diamond nanoparticle	Polysaccharide	[89]
	Skin repair	Sericin	Protein	[90,92]
	Wound dressing	Povidone-iodine, Polyhexamethylene biguanide (PHMB), Octenidine	Antiseptics	[60,98,131]
		Silver sulfadiazine	Antibacterial agent	[132]
		AuNPs	Nanoparticle	[100]
		AgNPs and antibiotics	Antibiotic	[133]
		Amoxicillin	Antibiotic	[134]
		Ceftriaxone	Antibiotic	[135]
		Curcumin and Lignin	Antibacterial agent	[136]
BNC	Skin repair	Vaccarin	Glycoside	[32,137,138]
	Diagnostics	Bovine serum albumin	Protein	[102,139]
	Tissue engineering	Theophylline	Drug/stimulant	[140]
	Wound healing	Paracetamol	Drug	[141]
	Tissue engineering	Lidocaine, ibuprofen	Local anesthetic	[142]
		Caffeine	Stimulant	[143]
		Berberine hydrochloride, Berberine sulfate	Drug	[144]
	Skin repair	Glycerin	Moisturizer	[145]
		Vanillin	Phenol	[146]
	Skin repair/wound healing	Hemoglobin, myoglobin, albumin, lysozyme	Proteins	[147]
	Tissue engineering	Glutamate decarboxylase	Enzyme	[148]
	Wound healing	Macrophage-stimulating protein	Protein	[94]
		Chitosan	Antibacterial	[89]
		Lidocaine	Anesthesia	[149]
	Skin repair	Glycerin	Moisturizer	[145]
		Chitin	Polysaccharide	[150]

**Table 3 ijms-26-01449-t003:** Cytotoxicity of nanocellulose-based biomaterials for tissue engineering, wound healing, and skin tissue repair.

Nanocellulose	Test	Cell Type/Organism	Toxicity Evaluation	Reference(s)
CNCs	Acute lethal test	Hepatocytes of fishes, in vitro rainbow trout hepatocyte assay	Low toxicity potential	[167]
	Multi-trophic assays		Low environmental risk	
	Cytotoxicity to human epithelial airways barrier	Monocyte-derived macrophages, dendritic cells, and bronchial epithelial cells	Low cytotoxicity, pro-inflammatory, cytokine production	[124]
	Skin irritation and sensitization tests	L929 cells	Low cytotoxicity	[168]
	Cytotoxicity		No cytotoxicity up to 48 h	
	MTT assayLDH assay	HBMEC, bEnd.3, RAW 264.7, MCF-10A, MDA-MB-231, MDA-MB-468, KB, PC-3 and C6	No cytotoxicity or inflammation	[169]
CNFs	Acute environmental toxicity, cytotoxicity	Human monocyte and mouse macrophages, Kinetic luminescent bacteria	No evidence of inflammatory effects or cytotoxicity	[162,170]
	Neurotoxicity and systemic effects	Nematode model	Low cytotoxicity	[33]
	In vitro pharyngeal aspiration study for pulmonary immunotoxicity and genotoxicity	Mice	Pulmonary inflammation, no DNA or chromosome damage	[171]
	Cytotoxicity, test of cell membrane, cell mitochondrial activity, DNA proliferation	3T3 fibroblast cells	Pure CNF: non-toxic, low cytotoxicity for CNF modified via PEI or CTAB surface modification	[123,172]
	Cytotoxicity	Bovine fibroblasts cells	Low cytotoxicity at low CNF concentration (0.02–100 µg/mL), no evidence of cytotoxicity for pure CNF, improved cytocompatibility of EPTMAC-modified CNF	[173]
BNC	Cytotoxicity	Osteoblast cells, L929 fibroblast cells	No evidence of cytotoxicity	[109,137]
	Cytotoxicity	Human umbilical vein endothelial cells	No evidence of toxicity in vitro and in vivo	[174]
	Cytotoxicity	C57/Bl6 male mouse	Non-toxic, non-immunogenic	[32]
	In vitro immunoreactivity, Cytotoxicity	Human umbilical vein endothelial cells		

PEI: polyethyleneimine; CTAB: cetyltrimethylammonium bromide; EPTMAC: epoxy propyl trimethylammonium chloride.

**Table 4 ijms-26-01449-t004:** Skin tissue engineering applications of nanocellulose-based composites.

Nanocellulose Type	Reinforcement Material	Synthesis Method	Enhanced Features	Application	Reference(s)
CNCs	Polyvinyl alcohol (PVA)/NCC scaffolds	Freeze-drying	Mechanical, thermal, and swelling features, uniform pore size	Cell adherence, growth, and metabolism	[120,159,249]
	Polylactide-polyglycolide (PLGA)	Electrospinning	Stretching and formation of fibrous types of scaffolds	Cell adhesion and fibroblast proliferation	
	Cellulose acetate propionate (CAP	Electrospinning and magnetic field	Small diameter	Development of blood vessels	[250]
	Coumarin and curcumin	Emulsion	Antioxidant, anti-inflammatory, antimicrobial, and anticancer activities	Wound dressings	[251,252]
	Polyvinyl alcohol/polyethylene oxide/CMC cellulose matrix	Electrospun	Antimicrobial/therapeutic	Wound dressings	[253,254]
CNFs	Brown algae nanofibrillar cellulose (BANFC)/ quaternized chitin/organic rectorite	Freeze-drying	Antibacterial activity and mechanical strength	Collagen formation and neovascularization	[255,256]
	Butylene succinate (PBS) and poly lactic acid (PLA)	Electron spun	Betterment of fiber structure, tensile strength, elastic modulus, and biocompatibility	Blood vessels	[257]
	Charged cellulose nanofibrils (cCNFs)	Polymer coating	Cell survival, adhesion, and proliferation	Artificial skin constructs	[147]
	Ca^2+^-crosslinked		Topical drug delivery	Chronic wound healing	[247]
	Copper-containing mesoporous bioactive glass		Antibacterial activity, angiogenic activity	Wound healing	[108]
	Gelatin and aminated silver nanoparticles		Mechanical andself-recovery properties, antibacterial activity	Skin wounds	[178]
BNC	Chitosan		Cell adhesion	Adhesion of human keratinocytes	[170,258]
	Keratin		Cell adhesion, proliferation, and morphology	Human skin keratinocytes, human skin fibroblasts	[95]
	Gelatin		Cell adhesion and proliferation	HaCaT line keratinocytes, wound closure efficacy	[226]
	Polypyrrole and polyaniline		Cell proliferation	Skin tissue engineering	[259]
	Paraffin microspheres		Support and cell proliferation	Mouse embryonic NIH 3T3 fibroblasts/tissue engineering	[57]

## Data Availability

All data of this study are available within the paper.

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
