# Peer review of "Current Paradigms and Future Challenges in Harnessing Nanocellulose for Advanced Applications in Tissue Engineering: A Critical State-of-the-Art Review for Biomedicine"

_ijms, 2025, doi:10.3390/ijms26041449_

Round 1
Reviewer 1 Report
Comments and Suggestions for Authors
Dear authors,
congratulations on your work! I am very happy to find a review article with such a detailed bibliometric section.
I have just two simple comments, almost irrelevant:
1) The first sentence of the section "Methodology and bibliometric analysis - When we performed a comprehensive literature survey on the Web of Science core collection, PubMed, ...” as written, seems to make no sense since nothing happens after you performed this comprehensive literature search. It seems to me that you meant that you affirmatively performed a comprehensive literature search on these bases. That's right, isn't it?
2) In teh section "Overview of the nanocellulose resources", I suggest reviewing the descriptions of the microorganisms which have mentioned the authors of the first description to include as much as possible the date of description, for example, Halocynthia roretzi (Drasche, 1884).
Best regards!
Author Response
Response to Reviewer #1
Comment 1: Dear authors, congratulations on your work! I am very happy to find a review article with such a detailed bibliometric section. I have just two simple comments, almost irrelevant:
Response: We are delighted to receive such a positive reference and are thankful to the reviewer for the kind evaluation of our manuscript. We sincerely appreciate the reviewer for encouragement that motivate us to continue our efforts in maintaining high quality research.
Comment 2: The first sentence of the section "Methodology and bibliometric analysis - When we performed a comprehensive literature survey on the Web of Science core collection, PubMed, ...” as written, seems to make no sense since nothing happens after you performed this comprehensive literature search. It seems to me that you meant that you affirmatively performed a comprehensive literature search on these bases. That's right, isn't it?
Response: Thank you for your keen insights. Yes, we wanted to infer that we have conducted a comprehensive literature survey on different scientific databases. The statement is revised on Lines 89-92 as follows:
Lines 89-92:
To compile this study, we performed a comprehensive literature survey on the Web of Science core collection, PubMed, ScienceDirect, and Google Scholar databases by entering keywords such as “Nanocellulose AND Tissue engineering” AND “Nanocellulose AND wound healing” AND “Nanocellulose and skin tissue repair”.
Comment 3: In the section "Overview of the nanocellulose resources", I suggest reviewing the descriptions of the microorganisms which have mentioned the authors of the first description to include as much as possible the date of description, for example, Halocynthia roretzi (Drasche, 1884).
Response: Thank you for this suggestion. We have now reviewed and updated the descriptions of microorganisms in the section "Overview of Nanocellulose Resources" to include the original authors and the date of their first description wherever possible. Please check the updated version of the manuscript for the following revision.
Lines 253-272:
Nanocellulose acquired from animals is pivotal for industry and biotechnology. Animal sources of nanocellulose are relatively limited compared to plant and bacterial sources. Among animals, tunicates like Styela clava (Herdman, 1881) [65] and Halocynthia roretzi (Drasche, 1884) [66] are widely known to produce significant amounts of cellulose that can be extracted and used for nanocellulose production. The cellulose content in tunicate tissues can be as high as 88% by weight, making them a rich source of cellulose [67]. The cellulose extracted from tunicates exhibits unique properties, such as high crystallinity and purity, making it suitable for producing nanocellulose, particularly CNC [68]. Nanocellulose derived from tunicates has been explored for various applications, including water treatment, biomedical engineering, and the development of nanocomposites [69]. Cellulose films produced from S. clava have been investigated for wound dressings [65]. They might also be used for sewing fibers, tissue engineering scaffolds, absorbable hemostats, and hemodialysis membranes [70]. The nanocellulose membrane extracted from the Halocynthia roretzi has been used for the removal of oils from wastewater [71]. Zhao and Li, investigated the tunicate-derived cellulose from four different species such as Ciona intestinalis (Linnaeus, 1767), Ascidia sp. (Linnaeus, 1767), Halocynthia roretzi, and Styela plicata (Lesueur, 1823) [72]. The authors compared the quantity, purity, and structural characteristics of cellulose extracted from these species and revealed that hard tunics obtained from Halocynthia roretzi, and Styela plicata yielded more cellulose than other species with 100% purity and 95% crystallinity.
References added:
[68] Lv, X.; Han, J.; Liu, M.; Yu, H; Liu K; Yang, Y; Sun, Y; Pan, P; Liang, Z; Chang, L; Chen, J Overview of preparation modification and application of tunicate-derived nanocellulose. Chemical Engineering Journal 2023, 452, 139439. https://doi.org/10.1016/j.cej.2022.139439
[72] Zhao, Y.; Li, J. Excellent chemical and material cellulose from tunicates: diversity in cellulose production yield and chemical and morphological structures from different tunicate species. Cellulose 2014, 21, 3427–3441. https://doi.org/10.1007/s10570-014-0348-619
Reviewer 2 Report
Comments and Suggestions for Authors
The development of biocompatible nanomaterials has emerged as an important research topic in recent years. Among them, nanocellulose has received significant research interest due to its unique properties such as biodegradability and biocompatibility. In this paper, the authors give a comprehensive review of the state-of-art on the application of nanocellulose in tissue engineering. With an extensive literature survey, they provide an in-depth analysis of the research data, including the discussion of the characterization, resources and preparation, biomedical applications of nanocellulose in tissue engineering. This paper not only highlights the importance of nanocellulose in biomedicine, but also underlines the hurdles and challenges of current research, providing insightful strategic directions for future research in this area. I would recommend the acceptance of this paper after making the following editions.
1. Regarding the limitations of nanocellulose, biosafety is still a key concern for the application in biomedicine, the authors should add more discussion in this section, which can refer to a recent review paper: Yan Li, Haixin Jiao, Hongxing Zhang, Xiangyu Wang, Yinyi Fu, Qianqian Wang, Huan Liu, Yang-chun Yong, Jiaqi Guo, Jun Liu,
Biosafety consideration of nanocellulose in biomedical applications: A review,
International Journal of Biological Macromolecules, Volume 265, Part 1, 2024, 130900,
https://doi.org/10.1016/j.ijbiomac.2024.130900.
2. Add some discussion on the surface modification/surface chemistry on nanocellulose, for this, a recent reference can be found: Dey, S., Singh, P., Sarkar, S. and Chowdhury, A.A. (2024). Applications of Nanocellulose in Tissue Engineering and Tissue Grafting. In Nanocellulose (eds M. Mukhopadhyay and D. Bhattacharya). https://doi.org/10.1002/9781394172825.ch7
3. Minor issue: Figure 4, the two dashed arrows need to be changed to solid arrows.
4. A related review paper on nanocellulose-based hybrid scaffolds needs to be added: Nanocellulose-Based Hybrid Scaffolds for Skin and Bone Tissue Engineering: A 10-Year Overview. Mridula Sreedharan, Raji Vijayamma, Elena Liyaskina, Viktor V. Revin, Muhammad Wajid Ullah, Zhijun Shi, Guang Yang, Yves Grohens, Nandakumar Kalarikkal, Khalid Ali Khan, and Sabu Thomas. Biomacromolecules 2024 25 (4), 2136-2155. DOI: 10.1021/acs.biomac.3c00975
Author Response
Response to Reviewer #2
Comment 1: The development of biocompatible nanomaterials has emerged as an important research topic in recent years. Among them, nanocellulose has received significant research interest due to its unique properties such as biodegradability and biocompatibility. In this paper, the authors give a comprehensive review of the state-of-art on the application of nanocellulose in tissue engineering. With an extensive literature survey, they provide an in-depth analysis of the research data, including the discussion of the characterization, resources and preparation, biomedical applications of nanocellulose in tissue engineering. This paper not only highlights the importance of nanocellulose in biomedicine, but also underlines the hurdles and challenges of current research, providing insightful strategic directions for future research in this area. I would recommend the acceptance of this paper after making the following editions.
Response: We sincerely appreciate the referee for his/her keen insights and valuable comments that helped us to improve the quality of our manuscript. Following the reviewer’s comments, we have incorporated the required sections and made significant revisions to the manuscript. We very much hope that the stage is now set more properly and the revised version of our paper will be welcomed by the referee.
Comment 2: Regarding the limitations of nanocellulose, biosafety is still a key concern for the application in biomedicine, the authors should add more discussion in this section, which can refer to a recent review paper: Yan Li, Haixin Jiao, Hongxing Zhang, Xiangyu Wang, Yinyi Fu, Qianqian Wang, Huan Liu, Yang-chun Yong, Jiaqi Guo, Jun Liu, Biosafety consideration of nanocellulose in biomedical applications: A review, International Journal of Biological Macromolecules, Volume 265, Part 1, 2024, 130900, https://doi.org/10.1016/j.ijbiomac.2024.130900
Response: Thank you for highlighting this critical aspect. A new section with the subheading, “Biosafety considerations of nanocellulose-based biomaterials” is added to the revised manuscript. This section highlights the biosafety concerns of nanocellulose for tissue engineering applications by incorporating insights from the recommended references and other reports. Please check the Lines 946-965 of the revised manuscript, which read as follows.
Lines 946-965:
Biosafety considerations of nanocellulose-based biomaterials
Although nanocellulose exhibits promising biocompatibility and non-toxicity to tissues, comprehensive biosafety evaluations are necessary before its widespread adoption in tissue engineering. A primary concern is the presence of endogenous and exogenous impurities, including residual lignin, hemicellulose, (1,3)-β-D-glucan, heavy metals, and bacterial endotoxins, which can provoke cytotoxic, immunogenic, or pro-inflammatory responses if not meticulously removed [178, 266-269]. Endotoxin contamination, particularly in BNC, necessitates stringent purification strategies such as alkali treatments, enzymatic detoxification, and high-temperature sterilization to meet pharmacopoeial standards [270]. Although, some in vitro and in vivo studies broadly support nanocellulose's biocompatibility, surface modifications including cationization can induce cytotoxic or genotoxic effects, necessitating precise control over functionalization [271]. Its interaction with the immune system and blood components further underscores the importance of hemocompatibility assessments, particularly in implantable and injectable applications [272]. Despite its promise of nanocellulose-based biomaterials for tissue engineering, the lack of standardized safety protocols across global regulatory frameworks remains a bottleneck for its clinical adoption [267]. Thus, future research must prioritize the development of universally accepted purification methodologies, long-term in vivo toxicity assessments, and mechanistic insights into nanocellulose’s biological fate to unlock its full potential as a next-generation biomaterial in regenerative medicine.
References added:
[266] Brião, G.V.; Rosa, D.S.; Frollini, E. Hydrogels from non-woody lignocellulosic biomass for toxic metal uptake from wastewater: A brief overview. Cellulose 2025, 32, 691–712. https://doi.org/10.1007/s10570-024-06321-w
[267] Li, Y.; Jiao, H.; Zhang, H.; Wang, X.; Fu, Y.; Wang, Q.; Liu, H.; Yong, Y.C.; Guo, J.; Liu, J. Biosafety consideration of nanocellulose in biomedical applications: A review. International journal of biological macromolecules, 2024, 265, 130900. https://doi.org/10.1016/j.ijbiomac.2024.130900
[268] Kaur, J.; Sengupta, P.; Mukhopadhyay, S. Critical Review of Bioadsorption on Modified Cellulose and Removal of Divalent Heavy Metals (Cd, Pb and Cu). Industrial & Engineering Chemistry Research 2022, 61(5), 1921–1954. https://doi.org/10.1021/acs.iecr.1c04583
[269] Zhang, C.; Liu, Q.; He, H.; Guo, X.; Yang, S.; Zhang, L.; Wang, L. Metal removal from heavy metal-enriched plants by deep eutectic solvents and its mechanism investigation.Separation and Purification Technology 2025, 357, 130189. https://doi.org/10.1016/j.seppur.2024.13018918
[270] Zuber, J.; Cascabulho, P.L.; Piperni, S.G.; do Amaral, R.J.; Vogt, C.; Carre, V.; Hertzog, J.; Kontturi, E.; Trubetskaya, A. Fast Easy and Reproducible Fingerprint Methods for Endotoxin Characterization in Nanocellulose and Alginate-Based Hydrogel Scaffolds. Biomacromolecules 2024, 25(10), 6762–6772. https://doi.org/10.1021/acs.biomac.4c00989
[271] Negi, A. Cationized Cellulose Materials: Enhancing Surface Adsorption Properties Towards Synthetic and Natural Dyes. Polymers 2025, 17, 36. https://doi.org/10.3390 /polym17010036
[272] Madani, M.; Borandeh, S.; Teotia, A.K.; Seppälä, J.V. Direct and Indirect Cationization of Cellulose Nanocrystals: Structure–Properties Relationship and Virus Capture Activity. Biomacromolecules 2023, 24(10), 4397–4407. https://doi.org/10.1021/acs.biomac.2c01045
Comment 3: Add some discussion on the surface modification/surface chemistry on nanocellulose, for this, a recent reference can be found: Dey, S., Singh, P., Sarkar, S. and Chowdhury, A.A. (2024). Applications of Nanocellulose in Tissue Engineering and Tissue Grafting. In Nanocellulose (eds M. Mukhopadhyay and D. Bhattacharya). https://doi.org/10.1002/9781394172825.ch7
Response: We appreciate the referee for bringing this point to our attention. We have now included a brief discussion on the surface modifications/functionalization of nanocellulose in the revised manuscript. Please read the Lines 578-617 of the revised manuscript.
Lines 578-617:
Surface modifications of nanocellulose
Nanocellulose's exceptional versatility stems from its ability to undergo tailored surface modifications, transforming it into a highly sophisticated and intelligent biomaterial for tissue engineering [187,188]. Hitherto, several reactions have been explored for the surface modification of nanocellulose such as esterification, sulfonation, phosphorylation, cationization, silanization, initiation, adsorption, coating, oxidation, polymer grafting, etc [189,190]. In particular chemical modification strategy-based functionalization enables precise control over its surface chemistry. By introducing functional groups like carboxyl, hydroxyl, or amino moieties through oxidation, coupling, or coating techniques, nanocellulose achieves enhanced biocompatibility, reactivity, and targeted bioactivity [191]. In tissue engineering applications, the functionalization of nanocellulose serves as an advanced platform for drug delivery, wound healing, and scaffold engineering, where its tailored surface chemistry enables antimicrobial activity, growth factor immobilization, and controlled drug release [192]. Moreover, these modifications enhance its compatibility with biological tissues, promote cell adhesion, and improve its interaction with bioactive molecules. For instance, TEMPO-mediated oxidation generates carboxyl-functionalized nanocellulose, facilitating crosslinking with polymers and proteins, thereby improving hydrogel stability and bioactivity [193]. Esterification and etherification impart hydrophobicity and mechanical resilience, making them ideal for scaffold reinforcement, while silanization enhances the interfacial properties of nanocellulose by introducing reactive silane groups for better integration with biomolecules.
Another strategy, phosphorylation of nanocellulose is explored for its bioactivity, making it highly relevant for bone tissue engineering applications [194]. By introducing phosphate groups onto the nanocellulose surface, this functionalization significantly increases the material’s affinity for calcium ions—an essential component of bone mineralization [191]. This property not only facilitates the formation of a mineralized matrix but also actively supports cellular adhesion, proliferation, and osteogenic differentiation, thereby improving the integration of engineered scaffolds with native bone tissue [195]. In the context of biomimetic bone regeneration, phosphorylated nanocellulose-based hydrogels could serve as dynamic scaffolds for calcium phosphate deposition, closely mimicking the natural biomineralization process crucial for skeletal tissue development [192]. Recent advancements have well underscored the potential of phosphorylation to nanocellulosic biomaterials [191, 196]. Wang et al. demonstrated the incorporation of phosphorylated-CNF into a dextran/methacrylated gelatin emulsion bioink for extrusion-based 3D bioprinting [196]. The modified bioink exhibited superior rheological behaviour, enhanced damping capacity, and increased mineralization ability, leading to improved cell viability, osteogenic differentiation, and biomineralized nodule formation. In summary, the surface modification of nanocellulose highlights its transformative potential in the development of next-generation bioactive scaffolds, offering a compelling route towards advanced regenerative therapies in orthopedic and craniofacial tissue engineering.
References added:
[187] Jeon, M.J.; Randhawa, A.; Kim, H.; Dutta, S.D.; Ganguly, K.; Patil, T.V.; Lee, J.; Acharya, R.; Park, H.; Seol, Y.; Lim, K.-T. Electroconductive Nanocellulose: A Versatile Hydrogel Platform for Biomedical Engineering Applications. Advanced Healthcare Materials 2025, 14, 2403983. https://doi.org/10.1002/adhm.202403983
[188] Dey, S.; Singh, P.; Sarkar, S.; Chowdhury, A.A. Applications of Nanocellulose in Tissue Engineering and Tissue Grafting. In Nanocellulose; Mukhopadhyay, M., Bhattacharya, D., Eds.; 2024, pp. 159–191. https://doi.org/10.1002/9781394172825.ch7
[189] Tortorella, S.; Buratti, V.V.; Maturi, M.; Sambri, L.; Franchini, M.C.; Locatelli, E. Surface-Modified Nanocellulose for Application in Biomedical Engineering and Nanomedicine: A Review. International Journal of Nanomedicine 2020, 15, 9909–9937. https://doi.org/10.2147/IJN.S26610316
[190] Sreedharan, M; Vijayamma R; Liyaskina E; Revin VV; Ullah MW; Shi Z; Yang G; Grohens Y; Kalarikkal N; Khan KA; Thomas S. Nanocellulose-Based Hybrid Scaffolds for Skin and Bone Tissue Engineering: A 10-Year Overview. Biomacromolecules 2024, 25(4), 2136–2155. https://doi.org/10.1021/acs.biomac.3c0097514
[191] Tamo, A.K. Nanocellulose-based hydrogels as versatile materials with interesting functional properties for tissue engineering applications. J. Mater. Chem.B 2024, 12, 7692–7759. https://doi.org/10.1039 /D4TB00397G15
[192] Kassie, B.B.; Getahun, M.J.; Azanaw, A.; Ferede, B.T.; Tassew, D.F. Surface modification of cellulose nanocrystals for biomedical and personal hygiene applications. International Journal of Biological Macromolecules 2024, 282, 136949. https://doi.org/10.1016/j.ijbiomac.2024.136949
[193] Ghasemlou, M.; Daver, F.; Ivanova, E.P.; Habibi, Y.; Adhikari, B. Surface modifications of nanocellulose: From synthesis to high-performance nanocomposites. Progress in Polymer Science 2021, 119, 101418. https://doi.org/10.1016/j.progpolymsci.2021.101418
[194] Patoary, M.K.; Islam, S.R.; Farooq, A.; Rashid, M.A.; Sarker, S.; Hossain, M.Y.; Rakib, M.A.N.; Al-Amin, M.; Liu, L. Phosphorylation of Nanocellulose: State of the Art and Prospects. Ind. Crops Prod. 2023, 201, 116965. https://doi.org/10.1016/j.indcrop.2023.11696513
[195] Zheng, Y.; Wang, L.; Bai, X.; Xiao, Y.; Che, J. Bio-inspired composite by hydroxyapatite mineralization on (bis)phosphonate-modified cellulose-alginate scaffold for bone tissue engineering. Colloids and Surfaces A: Physicochemical and Engineering Aspects 2022, 635, 127958. https://doi.org/10.1016 /j.colsurfa.2021.12795820
[196] Wang, Q.; Karadas, Ö.; Rosenholm, J.M.; Xu, C.; Näreoja, T.; Wang, X. Bioprinting Macroporous Hydrogel with Aqueous Two-Phase Emulsion-Based Bioink: In Vitro Mineralization and Differentiation Empowered by Phosphorylated Cellulose Nanofibrils. Advanced Functional Materials 2024, 34(29), 2400431. https://doi.org/10.1002/adfm.20240043117
Minor issue:
Comment 4: Figure 4, the two dashed arrows need to be changed to solid arrows.
Response: Thank you. Following the referee’s suggestion, we have updated the Figure 4 of the revised manuscript.
Comment 5: A related review paper on nanocellulose-based hybrid scaffolds needs to be added: Nanocellulose-Based Hybrid Scaffolds for Skin and Bone Tissue Engineering: A 10-Year Overview. Mridula Sreedharan, Raji Vijayamma, Elena Liyaskina, Viktor V. Revin, Muhammad Wajid Ullah, Zhijun Shi, Guang Yang, Yves Grohens, Nandakumar Kalarikkal, Khalid Ali Khan, and Sabu Thomas. Biomacromolecules 2024 25 (4), 2136-2155. DOI: 10.1021/acs.biomac.3c00975
Response: Thank you. The recommended reference is also cited in the revised manuscript on Line 584.
Reviewer 3 Report
Comments and Suggestions for Authors
This article provided a comprehensive review of nanocellulose materials for biomedical applications including tissue engineering, wound healing and drug delivery.
It began with the bibliometric analysis of cellulosic nanomaterials for above-mentioned applications as reported in the literature, followed by the introduction of nanocellulose structures such ad CNFs, CNCs, nanowhiskers, etc., and their potential applications in biomedical field. Moreover, the material sources, fabrication processes, and key challenges were also addressed and fully discussed. Based on these viewpoints, I recommended the publication of this article after minor revision by correcting some typographic errors as mentioned below,
There are few typographic errors in the references starting from Ref. 191 to Ref. 230. For example, 191.[191], 192.[192], 193.[193],.....etc.
Author Response
Response to Reviewer #3
Comment 1: This article provided a comprehensive review of nanocellulose materials for biomedical applications including tissue engineering, wound healing and drug delivery. It began with the bibliometric analysis of cellulosic nanomaterials for the above-mentioned applications as reported in the literature, followed by the introduction of nanocellulose structures such as CNFs, CNCs, nanowhiskers, etc., and their potential applications in biomedical field. Moreover, the material sources, fabrication processes, and key challenges were also addressed and fully discussed. Based on these viewpoints, I recommended the publication of this article after minor revision by correcting some typographic errors as mentioned below,
Response: Thank you for your kind recommendation and suggestions.
Comment 2: There are few typographic errors in the references starting from Ref. 191 to Ref. 230. For example, 191.[191], 192.[192], 193.[193], ..... etc.
Response: Thank you for pointing out the typographic errors. The formatting and typographic errors are corrected in the revised manuscript.
In addition to the comments of the Editor/reviewers, all word-spelling and grammatical errors were corrected in the revised version which makes it more appealing in its readability.
We look forward to hearing from you in due time regarding our submission. We would be glad to respond to any further questions and comments you may have.
Thanking you,
Sincerely yours,
Dr. Rongrong Xie